# GNN is a Counter?
# Revisiting GNN for Question Answering

**Kuan Wang**[1][*], **Yuyu Zhang** [1], **Diyi Yang**[1], **Le Song**[3,4] **& Tao Qin**[2]
[1]Georgia Institute of Technology   [2]Microsoft Research Asia   [3]BioMap   [4]MBZUAI
{kuanwang, yuyu, dyang888}@gatech.edu
lsong@cc.gatech.edu
taoqin@microsoft.com

## Abstract

Question Answering (QA) has been a long-standing research topic in AI and NLP fields, and a wealth of studies have been conducted to attempt to equip QA systems with human-level reasoning capability. To approximate the complicated human reasoning process, state-of-the-art QA systems commonly use pre-trained language models (LMs) to access knowledge encoded in LMs together with elaborately designed modules based on Graph Neural Networks (GNNs) to perform reasoning over knowledge graphs (KGs). However, many problems remain open regarding the reasoning functionality of these GNN-based modules. Can these GNN-based modules really perform a complex reasoning process? Are they under- or over-complicated for QA? To open the black box of GNN and investigate these problems, we dissect state-of-the-art GNN modules for QA and analyze their reasoning capability. We discover that even a very simple graph neural counter can outperform all the existing GNN modules on *CommonsenseQA* and *OpenBookQA*, two popular QA benchmark datasets which heavily rely on knowledge-aware reasoning. Our work reveals that existing knowledge-aware GNN modules may only carry out some simple reasoning such as counting. It remains a challenging open problem to build comprehensive reasoning modules for knowledge-powered QA.

## 1 Introduction

Accessing and reasoning over relevant knowledge is the key to Question Answering (QA). Such knowledge can be implicitly encoded or explicitly stored in structured knowledge graphs (KGs). Large pre-trained language models (Devlin et al., 2018; Radford et al., 2018; 2019; Brown et al., 2020) are found to be effective in learning broad and rich implicit knowledge (Petroni et al., 2019; Bosselut et al., 2019; Talmor et al., 2020) and thus demonstrate much success for QA tasks. Nevertheless, pre-trained LMs struggle a lot with structured reasoning such as handling negation (Ribeiro et al., 2020; Yasunaga et al., 2021). In contrast, the explicit knowledge such as knowledge graphs (KGs) (Speer et al., 2017; Bollacker et al., 2008) works better for structured reasoning as it explicitly maintains specific information and relations and often produces interpretable results such as reasoning chains (Jhamtani & Clark, 2020; Khot et al., 2020; Clark et al., 2020b).

To utilize both implicit and explicit knowledge for QA, many existing works combine large pre-trained LMs with Graph Neural Networks (GNNs; Scarselli et al. (2008); Kipf & Welling (2017); Veličković et al. (2018)), which are shown to achieve prominent QA performance. These approaches commonly follow a two-step paradigm to process KGs: 1) *schema graph grounding* and 2) *graph modeling for inference*. In Step 1, a schema graph is a retrieved sub-graph of KG related to the QA context and grounded on concepts; such sub-graphs include nodes with concept text, edges with relation types, and their adjacency matrix. In Step 2, graph modeling is carried out via an elaborately designed graph-based neural module. For instance, Lin et al. (2019) uses GCN-LSTM-HPA which combines graph convolutional networks (Kipf & Welling, 2017) and LSTM (Hochreiter & Schmidhuber, 1997)

---

[*]Work done during an internship at MSRA

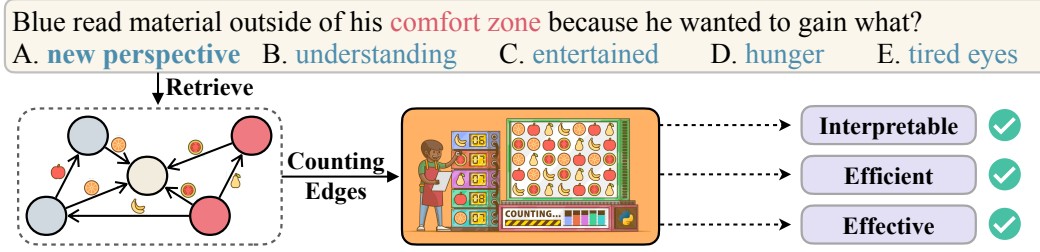

Figure 1: We analyze state-of-the-art GNN modules for the task of KG-powered question answering, and find that the counting of edges in the graph plays an essential role in knowledge-aware reasoning. Accordingly, we design an efficient, effective and interpretable graph neural counter module for knowledge-aware QA reasoning.

with a hierarchical path-based attention mechanism for path-based relational graph representation. Feng et al. (2020) extends the single-hop message passing of RGCN (Schlichtkrull et al., 2018) as multi-hop message passing with structured relational attention to obtain the path-level reasoning ability and intractability, while keeping the good scalability of GNN. Yasunaga et al. (2021) uses a LM to encode QA context as a node in the scheme graph and then utilized graph attention networks (Veličković et al., 2018) to process the joint graph.

Given that today's QA systems have become more and more complicated, we would like to revisit those systems and ask several basic questions: Are those GNN-based modules under- or over-complicated for QA? What is the essential role they play in reasoning over knowledge? To answer these questions, we first analyze current state-of-the-art GNN modules for QA and their reasoning capability. Building upon our analysis, we then design a simple yet effective graph-based neural counter that achieves even better QA performance on *CommonsenseQA* and *OpenBookQA*, two popular QA benchmark datasets which heavily rely on knowledge-aware reasoning.

In the analysis part, we employ Sparse Variational Dropout (SparseVD; Molchanov et al. (2017)) as a tool to dissect existing graph network architectures. SparseVD is proposed as a neural model pruning method in the literature, and its effect of model compressing serves as an indicator to figure out which part of the model can be pruned out without loss of accuracy. We apply SparseVD to the inner layers of GNN modules, using their sparse ratio to analyze each layer's contribution to the reasoning process. Surprisingly, we find that those GNN modules are over-parameterized: some layers in GNN can be pruned to a very low sparse ratio, and the initial node embeddings are dispensable.

Based on our observations, we design Graph Soft Counter (GSC), a very simple graph neural model which basically serves as a counter over the knowledge graph. The hidden dimension of GSC layers is only 1, thus each edge/node only has a single number as the hidden embedding for graph-based aggregation. As illustrated in Figure 1, GSC is not only very efficient but also interpretable, since the aggregation of 1-dimensional embedding can be viewed as soft counting of edge/node in graphs. Although GSC is designed to be a simplistic model, which has less than 1% trainable parameters compared to existing GNN modules for QA, it outperforms state-of-the-art GNN counterparts on two popular QA benchmark datasets. Our work reveals that the existing complex GNN modules may just perform some simple reasoning such as counting in knowledge-aware reasoning.

The key contributions of our work are summarized as follows:

- *Analysis of existing GNN modules:* We employ SparseVD as a diagnostic tool to analyze the importance of various parts of state-of-the-art knowledge-aware GNN modules. We find that those GNN modules are over-complicated for what they can accomplish in the QA reasoning process.

- *Importance of edge counting:* We demonstrate that the counting of edges in the graph plays a crucial role in knowledge-aware reasoning, since our experiments show that even a simple hard counting model can achieve QA performance comparable to state-of-the-art GNN-based methods.

- *Design of GSC module:* We propose Graph Soft Counter (GSC), a simple yet effective neural module as the replacement for existing complex GNN modules. With less than 1% trainable parameters compared to existing GNN modules for QA, our method even outperforms those complex GNN modules on two benchmark QA datasets.

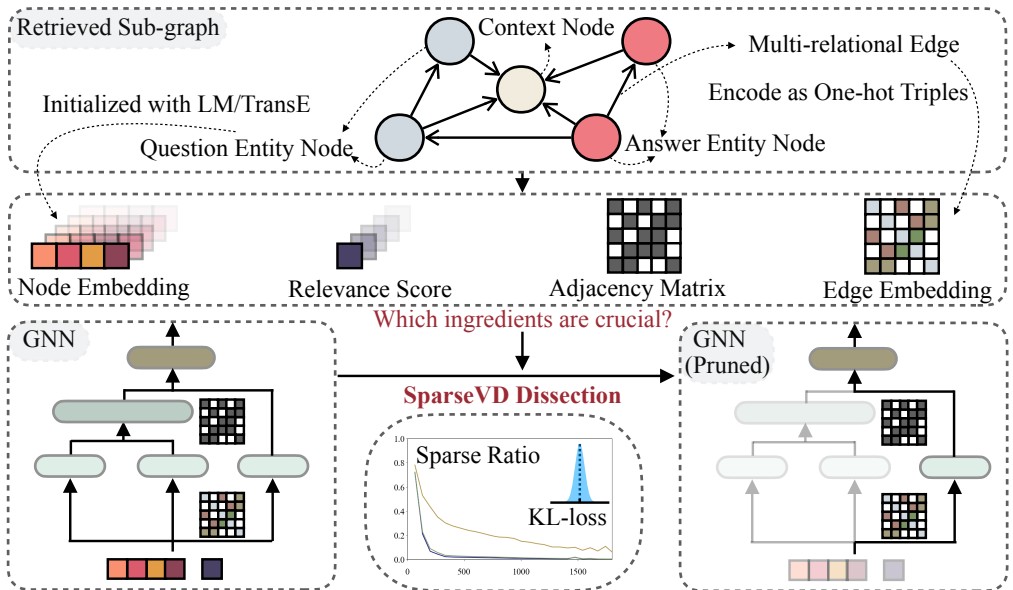

Figure 2: The retrieved sub-graph of KG is formulated as entity nodes representing concepts connected by edges representing relations and the central context node connected to all question entity nodes and answer entity nodes. The pre-processed graph data generally has the following ingredients: node embedding initialized with pre-trained KG embeddings, relevance score computed by LM, adjacency matrix representing the topological graph structure, edge embeddings to encode the node types, and edge information of relation types. We adapt SparseVD as a diagnostic tool to dissect GNN-based reasoning modules for QA, getting the sparse ratio of each layer to indicate its importance. We find that some layers and ingredients are completely dispensable, which inspires us to design a simple, efficient and effective GNN module as the replacement of existing complex GNN modules.

## 2 ANALYSIS

### 2.1 PRELIMINARIES

The knowledge required by QA systems typically comes from two sources: implicit knowledge in pre-trained language models and explicit knowledge in knowledge graphs.

To use the implicit knowledge, existing works commonly use LMs as the encoder to encode textual input sequence $\mathbf{x}$ to contextualized word embeddings, and then pool the embedding of the start token (e.g., $[CLS]$ for BERT) as the sentence embedding. In addition, a MLP (we use a one layer fully connected layer) is used to map the sentence embedding to the score for the choice.

To process the explicit knowledge in knowledge graphs, existing works commonly follow a two-step paradigm: *schema graph grounding* and *graph modeling for inference*. The schema graph is a retrieved sub-graph of KG grounding on concepts related to the QA context. We define the sub-graph as a multi-relational graph $\mathcal{G} = (\mathcal{V}, \mathcal{E})$, where $\mathcal{V}$ is the set of entity nodes (concepts) in the KG and $\mathcal{E} \subseteq \mathcal{V} \times \mathcal{R} \times \mathcal{V}$ is the set of triplet edges that connect nodes in $\mathcal{V}$ with relation types in $\mathcal{R}$. Following prior works (Lin et al., 2019; Yasunaga et al., 2021), we link the entities mentioned in the question $q$ and answer choice $a \in \mathcal{C}$ to the given sub-graph $\mathcal{G}$.

A wealth of existing works have explored to incorporate pre-trained language models with knowledge-aware graph neural modules. These methods usually have complex architecture design to process complex input knowledge. As shown in Figure 2, the processed retrieved sub-graph can be summarized as four main components:

**Node embeddings.** Many existing works on GNN-based QA employ external embeddings to initialize the node embeddings in the graph. For example, Lin et al. (2019) employs TransE (Wang et al., 2014) with GloVe embeddings (Pennington et al., 2014) as the initial node embeddings. Feng et al. (2020) and Yasunaga et al. (2021) use mean-pooled BERT embeddings of entities across the graph as the initialization of node embeddings.

**Relevance score.** Yasunaga et al. (2021) uses an extra embedding of relevance score in the input node features to estimate the importance of KG nodes relative to the given QA context. The relevance

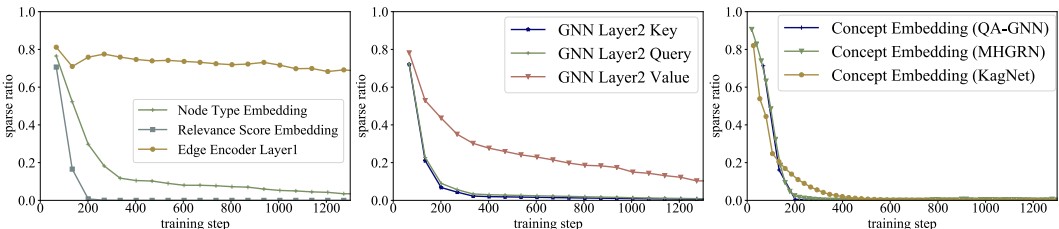

Figure 3: The sparse ratio curves obtained from the SparseVD training of GNN reasoning modules: 1) the left plot shows the curves for the embedding layers in QA-GNN, where the node score and the node type reach a low ratio, while the edge encoder preserves a large ratio; 2) the middle plot shows the curves for the layers within the QA-GNN graph attention layer, where the key and query layers converge to a fairly low ratio, while the value layer has a relatively large ratio; 3) the right plot shows the curves of the initial node embedding layers of three representative GNN-based QA methods, where all the ratios are close to 0, indicating that the initial node embeddings are dispensable.

score is computed by the masked LM loss of the stitched sequence with the QA context and the retrieved concept.

**Adjacency matrix.** The original adjacency matrix of the sub-graph is asymmetric. Typically, the adjacency matrix is converted to be symmetric before feeding into the GNN module: each KG relation is double recorded with opposite directions as symmetric edges in the processed adjacency matrix.

**Edge embeddings.** Existing works use various ways to obtain the edge embeddings. Yasunaga et al. (2021) uses concatenated one-hot vector $[u_s, e_{st}, u_t]$ to encode edge triplets, where $u_s, u_t$ indicates the node type and $e_{st}$ indicates the edge type. Lin et al. (2019) uses TransE (Wang et al., 2014) initialized with GloVe embeddings to generate edge embeddings. Feng et al. (2020) designs product-based edge embeddings to approximate the multi-hop relations in the graph.

## 2.2 DISSECTION

To investigate the mechanism of these complex systems and how they use the complex information, we introduce a neural model pruning method named Sparse Variational Dropout (SparseVD) (Molchanov et al., 2017) as a diagnostic tool to automatically dissect the graph network architecture. Note that our dissection tool is pruning method agnostic; other pruning schemes (Han et al., 2015; He et al., 2017; Liu et al., 2017) may also be applicable. Here we choose SparseVD since it prunes not only the weights with smaller scale but also the weights with higher variance, and it is theoretically supported by stochastic variational inference (Hoffman et al., 2013; Kingma & Welling, 2013; Rezende et al., 2014; Kingma et al., 2015).

Despite the fact that SparseVD was originally proposed in the field of model compression (Han et al., 2016; He et al., 2018; Lin et al., 2017; Zhou et al., 2018; Wang et al., 2018), we use it to investigate which parts of GNN can be pruned out (sparse ratio to zero) without loss of accuracy, which indicates that part of the model is redundant. To be specific, we keep the target model architecture unchanged, and parameterize each part of the weights in the model as a Gaussian distribution. Afterwards, this probabilistic model will be trained with a cross-entropy loss jointly with a KL-divergence regularization term. So the joint loss constrains the weights to our pruning prior. We implement the SparseVD with the default threshold as in Molchanov et al. (2017). Eventually, we get the pruned model with different sparsified ratios for the layers. As shown in Table 1, we investigate three representative GNN-based QA methods, and the sparsified models could achieve the accuracy of their original counterparts. This indicates that our dissection regularization does not hurt the model performance, so we then dive into each layer to see which parts play the most important role.

| Methods | w/o SparseVD | | w/ SparseVD | |
|---|---|---|---|---|
| | **IHdev-Acc.** (%) | **IHtest-Acc.** (%) | **IHdev-Acc.** (%) | **IHtest-Acc.** (%) |
| KagNet (Lin et al., 2019) | 73.47 ($\pm$0.22) | 69.01 ($\pm$0.76) | 75.18 ($\pm$1.05) | 70.48 ($\pm$0.77) |
| MHGRN (Feng et al., 2020) | 74.45 ($\pm$0.10) | 71.11 ($\pm$0.81) | 77.15 ($\pm$0.32) | 72.66 ($\pm$0.61) |
| QAGNN (Yasunaga et al., 2021) | 76.54 ($\pm$0.21) | 73.41 ($\pm$0.92) | 77.64 ($\pm$0.50) | 73.57 ($\pm$0.48) |

Table 1: To preserve the reasoning ability for analysis, our SparseVD tool prunes the GNN-based models without loss of accuracy on *CommonsenseQA* dataset. As the official test is hidden, here we report the in-house dev (IHdev) and test (IHtest) accuracy, following the data split of Lin et al. (2019).

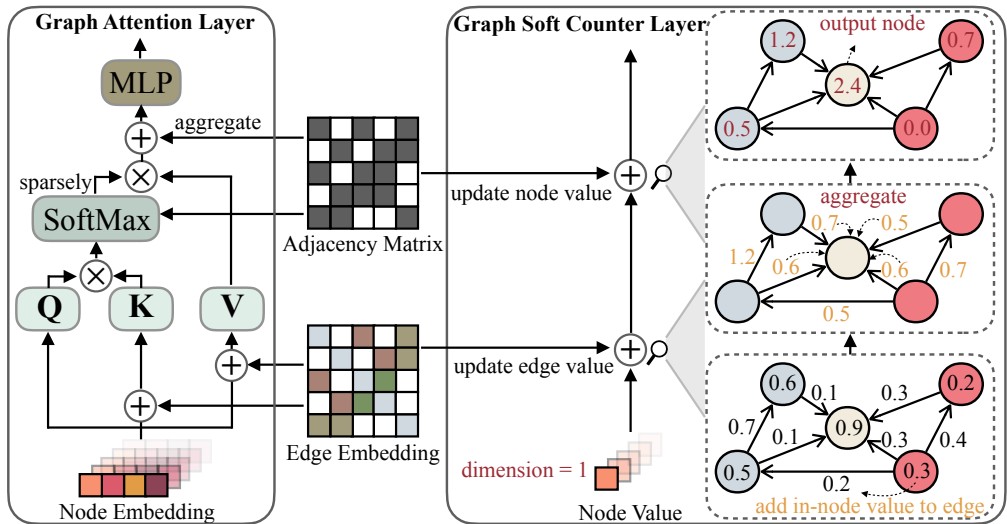

Figure 4: Graph Soft Counter (right) extremely simplifies the architecture of conventional GNNs (left). The GSC layers are parameter-free and only keep the basic graph operations: 1) update each edge embedding with incoming node (in-node) embeddings; 2) update each node embedding by aggregating the edge embeddings. Since we reduce the hidden dimension to only 1, GSC can be viewed as a soft counter over the graph for generating the final score of the output node.

## 2.3 OBSERVATIONS AND HYPOTHESIS

In Figure 3, we plot the sparse ratio of some representative layers during the SparseVD training of GNN reasoning modules. Due to the limited space, refer to Appendix A.3 for the full plots of curves. According to these plots, we summarize our key observations as follows: 1) the left plot shows that the edge encoder encoding edge triplets (edge / node type information) preserves a relatively higher sparse ratio, while the node score embedding layer can be fully pruned; 2) the middle plot shows the layers inside GNN, and all the sparse ratios are low while the value layer has a relatively higher sparse ratio than key/query layers; 3) the right plot shows the concept embedding layers of three representative GNN methods, which process the initial node embeddings, can be completely discarded. Inspired by these findings, we come up with the following design guidelines for a much simpler yet effective graph neural module for knowledge-aware reasoning:

**Node embeddings & relevance score.** We observe that both the initial node (concept) embeddings and the relevance score embeddings are dispensable in our experiments. Therefore, we may directly remove the initial node embedding layer and the relevance score embedding layer.

**Edge embeddings.** The edge encoder layers are hard to prune which indicates that edge/node type information is essential for reasoning, so we keep the edge embeddings.

**Message passing layers.** The linear layers inside GNN layers (query, key, value, etc) can be pruned to a very low sparse ratio, suggesting that GNN may be over-parameterized in these systems, and we can use fewer parameters for these layers.

**Graph pooler.** The final attention-based graph pooler aggregates node representations over the graph to get the graph representation. We observe that the key/query layers inside the pooler can be pruned out; as a result, the graph pooler can be reduced to a linear transformation.

**Algorithm 1** PyTorch-style code of GSC

```
# qa_context: question answer pair context
# adj: edge index with shape 2 x N_edge
# edge_type: edge type with shape 1 x N_edge
# node_type: node type with shape 1 x N_node
edge_emb = edge_encoder(adj, edge_type, node_type)
node_emb = torch.zeros(N_node)
for i_layer in range(num_gsc_layers):
    # propagate from node to edge
    edge_emb += node_emb[adj[1]]
    # aggregate from edge to node
    node_emb = scatter(edge_emb, adj[0])
graph_score = node_emb[0]
context_score = fc(roberta(qa_context))
qa_score = context_score + graph_score
```

## 3 GRAPH SOFT COUNTER

Based on our findings in Section 2, we design the Graph Soft Counter (GSC), a simplistic graph neural module to process the graph information. As demonstrated in Algorithm 1, there are only two

basic components in GSC to compute the graph score: Edge encoder and Graph Soft Counter layers. We can get the QA choice score by simply summing up the graph score and context score.

The edge encoder is a two-layer MLP with dimension $[46 \times 32 \times 1]$ followed by a Sigmoid function to encode the edge triplets to a float number in the range of $(0, 1)$. The triplets is represented as a concatenated one-hot vector $[u_s, e_{st}, u_t]$, where $u_s, u_t$ indicates 4 node types and $e_{st}$ indicates 38 relation types (17 regular relations plus question/answer entity relations and their reversions).

GSC layers are parameter-free and only keep the basic graph operations: Propagation and aggregation. To overcome over-parameterization, one straightforward way is to reduce the hidden size, and the extreme case is reducing it to only 1. As shown in Figure 4, the GSC layer simply propagates and aggregates the numbers on the edges and node following the two-step scheme: 1) update the edge value with in-node value, and this is done by simply indexing and adding; 2) update the node value by aggregating the edge values, and this is done by scattering edge values to out-node.

Since we reduce the hidden size to only 1, GSC is basically propagating and aggregating numbers on the graph. We can interpret these numbers as soft counts representing the importance of the edges and nodes. The aggregation of GSC can be regarded as accumulating the soft counts, so call it Graph Soft Counter. In addition, we can also formulate GSC as an extremely efficient variant of current mainstream GNN such as GAT and GCN. As shown in Table 3, the

| | KagNet | MHGRN | QA-GNN | GSC (ours) |
|---|---|---|---|---|
| Adjacency matrix | ✓ | ✓ | ✓ | ✓ |
| Edge types | ✓ | ✓ | ✓ | ✓ |
| Node types | × | ✓ | ✓ | ✓ |
| Node embeddings | ✓ | ✓ | ✓ | × |
| Relevance score | × | × | ✓ | × |
| #Learnable params | 700K | 547K | 2845K | **3K** |
| Model size | 819M | 819M | 821M | **3K** |

Table 2: Our GSC does not use initial node embeddings and relevance score as in previous works, making our model get rid of concept embedding parameters and thus its model size is extremely small. Since our model does not have any learnable parameters except the edge encoder, our model has much less chance to be over-parameterized.

| Model | Time | Space |
|---|---|---|
| | *$\mathcal{G}$ is a dense graph* | |
| $L$-hop KagNet | $\mathcal{O}\left(|\mathcal{R}|^L|\mathcal{V}|^{L+1}L\right)$ | $\mathcal{O}\left(|\mathcal{R}|^L|\mathcal{V}|^{L+1}L \cdot D\right)$ |
| $L$-hop MHGRN | $\mathcal{O}\left(|\mathcal{R}|^2|\mathcal{V}|^2L\right)$ | $\mathcal{O}\left(|\mathcal{R}||\mathcal{V}|L \cdot D\right)$ |
| $L$-layer QA-GNN | $\mathcal{O}\left(|\mathcal{V}|^2L\right)$ | $\mathcal{O}\left(|\mathcal{R}||\mathcal{V}|L \cdot D\right)$ |
| $L$-layer GSC | $\mathcal{O}\left(|\mathcal{V}|L\right)$ | $\mathcal{O}\left(|\mathcal{R}||\mathcal{V}|L\right)$ |
| | *$\mathcal{G}$ is a sparse graph with maximum node degree $\Delta \ll |\mathcal{V}|$* | |
| $L$-hop KagNet | $\mathcal{O}\left(|\mathcal{R}|^L|\mathcal{V}|L\Delta^L\right)$ | $\mathcal{O}\left(|\mathcal{R}|^L|\mathcal{V}|L\Delta^L \cdot D\right)$ |
| $L$-hop MHGRN | $\mathcal{O}\left(|\mathcal{R}|^2|\mathcal{V}|L\Delta\right)$ | $\mathcal{O}\left(|\mathcal{R}||\mathcal{V}|L \cdot D\right)$ |
| $L$-layer QA-GNN | $\mathcal{O}\left(|\mathcal{V}|L\Delta\right)$ | $\mathcal{O}\left(|\mathcal{R}||\mathcal{V}|L \cdot D\right)$ |
| $L$-layer GSC | $\mathcal{O}\left(|\mathcal{V}|L\right)$ | $\mathcal{O}\left(|\mathcal{R}||\mathcal{V}|L\right)$ |

Table 3: GSC is extremely efficient compared to the computation complexity of $L$-hop reasoning models with hidden dimension $D$ on a dense / sparse graph $\mathcal{G} = (\mathcal{V}, \mathcal{E})$ with the relation set $\mathcal{R}$.

computation complexity of GSC is much smaller than the baselines regard to both time and space. Table 2 shows that the trainable parameters in GSC is remarkably less than previous methods since our message-passing layers are parameter-free. The model size of GSC is also extremely small since we do not use any initial node embedding.

To draw a more conclusive conclusion, we handcraft a counting-based feature and use a simple two-layer MLP to map it to graph score. The feature is computed by counting the occurrence of each possible edge triplet. Surprisingly, this simple model could achieve similar performance and can even outperform baselines, which further proves that Counter is a basic and crucial part in QA.

## 4 EXPERIMENTS

### 4.1 SETTINGS

**Datasets and KG.** We conduct extensive experiments on *CommonsenseQA* (Talmor et al., 2019) and *OpenBookQA* (Mihaylov et al., 2018), two popular QA benchmark datasets that heavily rely on knowledge-aware reasoning capability. *CommonsenseQA* is a 5-way multiple choice QA task that requires reasoning with commonsense knowledge, which contains 12,102 questions. The test set of *CommonsenseQA* is not publicly available, and the model predictions can only be evaluated once every two weeks via the official leaderboard. Hence, following the data split of Lin et al. (2019), we experiment and report the accuracy on the in-house dev (IHdev) and test (IHtest) splits. We also report the accuracy of our final system on the official test set. *OpenBookQA* is a 4-way multiple choice QA task that requires reasoning with elementary science knowledge, which contains 5,957 questions. Following Clark et al. (2020a), methods with AristoRoBERTa use textual evidence as an external

input of the QA context. In addition, we use *ConceptNet* (Speer et al., 2017), a general commonsense knowledge graph, as our structured knowledge source $\mathcal{G}$ for both of the above tasks. Given each QA context (question and answer choice), we retrieve the sub-graph $\mathcal{G}_{sub}$ from $\mathcal{G}$ following Feng et al. (2020). We only use the concepts that occur in the QA context, as detailed in Table 8.

**Implementation and training details.** We use a two-layer MLP with hidden dimension $47 \times 32 \times 1$ followed by a Sigmoid function as our edge encoder, and the number of GSC layers is 2. Since GSC is extremely efficient, we set the dropout (Srivastava et al., 2014) rate to 0. We use RAdam (Liu et al., 2020) as the optimizer and set the batch size to 128. The learning rate is 1e-5 for RoBERTa and 1e-2 for GSC. The maximum number of epoch is set to 30 for *CommonsenseQA* and 75 for *OpenBookQA*. On a single Quadro RTX6000 GPU, each GSC training only takes about 2 hours to converge, while other methods often take 10+ hours.

**Baselines.** For GSC and all the other methods, RoBERTa-large (Liu et al., 2019) is used for both *CommonsenseQA* and *OpenBookQA*, and AristoRoBERTa (Clark et al., 2020a) is used for *OpenBookQA* as an additional setting for fair comparison. We experiment to compare GSC with existing GNN-based QA methods, including RN (Santoro et al., 2017), GconAttn (Wang et al., 2019), RGCN (Schlichtkrull et al., 2018), KagNet (Lin et al., 2019), MHGRN (Feng et al., 2020) and QA-GNN (Yasunaga et al., 2021), which only differ in the design of GNN reasoning modules. We report the performance of baselines referring to Yasunaga et al. (2021) and all the test results are evaluated on the best model on the dev split.

| Model | IHdev-Acc. (%) | IHtest-Acc. (%) |
|---|---|---|
| RoBERTa-large (w/o KG) | 73.07 ($\pm$0.45) | 68.69 ($\pm$0.56) |
| + RGCN (Schlichtkrull et al., 2018) | 72.69 ($\pm$0.19) | 68.41 ($\pm$0.66) |
| + GconAttn (Wang et al., 2019) | 72.61( $\pm$0.39) | 68.59 ($\pm$0.96) |
| + KagNet (Lin et al., 2019) | 73.47 ($\pm$0.22) | 69.01 ($\pm$0.76) |
| + RN (Santoro et al., 2017) | 74.57 ($\pm$0.91) | 69.08 ($\pm$0.21) |
| + MHGRN (Feng et al., 2020) | 74.45 ($\pm$0.10) | 71.11 ($\pm$0.81) |
| + QA-GNN (Yasunaga et al., 2021) | 76.54 ($\pm$0.21) | 73.41 ($\pm$0.92) |
| + GSC (Ours) | **79.11** ($\pm$0.22) | **74.48** ($\pm$0.41) |

Table 4: Performance comparison on *CommonsenseQA* in-house split (controlled experiments). As the official test is hidden, here we report the in-house dev (IHdev) and test (IHtest) accuracy, following the data split of Lin et al. (2019).

| Model | Test |
|---|---|
| RoBERTa (Liu et al., 2019) | 72.1 |
| RoBERTa + FreeLB (Zhu et al., 2020) (ensemble) | 73.1 |
| RoBERTa + HyKAS (Ma et al., 2019) | 73.2 |
| RoBERTa + KE (ensemble) | 73.3 |
| RoBERTa + KEDGN (ensemble) | 74.4 |
| XLNet + GraphReason (Lv et al., 2020) | 75.3 |
| RoBERTa + MHGRN (Feng et al., 2020) | 75.4 |
| ALBERT + PG (Wang et al., 2020) | 75.6 |
| RoBERTa + QA-GNN (Yasunaga et al., 2021) | 76.1 |
| ALBERT (Lan et al., 2019) (ensemble) | 76.5 |
| UnifiedQA (11B)[*] (Khashabi et al., 2020) | **79.1** |
| RoBERTa + GSC (Ours) | 76.4 |

Table 5: Test accuracy on *CommonsenseQA*'s official leaderboard. The existing top system, UnifiedQA (11B params) is 30x larger than our model.

## 4.2 RESULTS

***CommonsenseQA*.** As shown in Table 4, GSC outperforms the previous best model with $2.57\%$ mean accuracy on In-house dev split and $1.07\%$ mean accuracy on the in-house test split. We observe that the performance variance of GSC is smaller than the baselines, indicating that our GSC are both effective and stable. On the official leaderboard of *CommonsenseQA* in Table 5, GSC also outperforms all the GNN-based QA systems. Note that the previous top system UnifiedQA (11B params) uses T5 (Raffel et al., 2020) as the pre-trained LM model, which is 30x larger than our model and uses much more pre-training data.

| Model | RoBERTa-large | AristoRoBERTa |
|---|---|---|
| Fine-tuned LMs (w/o KG) | 64.80 ($\pm$2.37) | 78.40 ($\pm$1.64) |
| + RGCN | 62.45 ($\pm$1.57) | 74.60 ($\pm$2.53) |
| + GconAtten | 64.75 ($\pm$1.48) | 71.80 ($\pm$1.21) |
| + RN | 65.20 ($\pm$1.18) | 75.35 ($\pm$1.39) |
| + MHGRN | 66.85 ($\pm$1.19) | 80.6 |
| + QA-GNN | 67.80 ($\pm$2.75) | 82.77 ($\pm$1.56) |
| + GSC (Ours) | **70.33** ($\pm$0.81) | **86.67** ($\pm$0.46) |

Table 6: Test accuracy on *OpenBookQA*. Methods with AristoRoBERTa use the textual evidence by Clark et al. (2020a) as an additional input to the QA context.

***OpenBookQA*.** From Table 6 our GSC outperforms the previous best model with $2.53\%$ mean test accuracy with normal setting and $3.90\%$ test accuracy with the AristoRoBERTa setting. More remarkably, as shown in Table 7, our GSC ranks top one on the official leaderboard of *OpenBookQA*, which even surpasses the performance of UnifiedQA (11B), which is 30x larger than our model.



Figure 5: Venn diagrams for the prediction overlap of different models and ground truth (GT) of the IHtest split of CommonsenseQA. The ALBERT has the least overlap since all the left four use RoBERTa to encode the QA context. The order of the percentage of overlap of left four exactly matches the order of the performance of each models: GSC > QA-GNN > MHGRN > RoBERTa. This indicates that the reasoning capability learned by our GSC basically covers that of other GNNs.

## 4.3 DISCUSSION

As mentioned above, we observe that the initial node embeddings are dispensable. Then we start to explore how the maximum number of retrieved nodes (also related to edges) affects the model. We experiment various numbers of nodes for our GSC, and summarize the results in Table 8. We find that larger maximum number of node does not benefit the model, and the model achieves the best performance when we use all and only the entity nodes directly occur in question and answer, whose number of nodes is generally less than 32. This indicates that 1-hop retrieval is adequate for our methods, and this could be done super efficiently than multi-hop retrieval.

To further analyze the reasoning capacity of GSC, we draw the Venn diagram for the predictions of different methods and ground truth (GT) in Figure 5. We find that even for the different runs of the same GSC model, the correct overlap is only 69%, showing that the datasets are relatively noisy and there exists decent variance in the prediction results. We also observe that GSC has a larger overlap for GNN-based systems (e.g., QA-GNN and MHGRN), while at the same time having less overlap for non-GNN methods. The ALBERT model has the least overlap since all the other methods use RoBERTa as the text encoder.

| Model | Test |
|---|---|
| Careful Selection (Banerjee et al., 2019) | 72.0 |
| AristoRoBERTa | 77.8 |
| KF + SIR (Banerjee & Baral, 2020) | 80.0 |
| AristoRoBERTa + PG (Wang et al., 2020) | 80.2 |
| AristoRoBERTa + MHGRN (Feng et al., 2020) | 80.6 |
| ALBERT + KB | 81.0 |
| AristoRoBERTa + QA-GNN | 82.8 |
| T5[*] (Raffel et al., 2020) | 83.2 |
| UnifiedQA (11B)[*] (Khashabi et al., 2020) | 87.2 |
| AristoRoBERTa + GSC (Ours) | **87.4** |

Table 7: Test accuracy on *OpenBookQA* leaderboard. All listed methods use the provided science facts as an additional input to the language context. The previous top 2 systems, UnifiedQA (11B params) and T5 (3B params) are 30x and 8x larger than our model.

We also observe that the order of the percentage of overlap of left four exactly matches the order of the performance of each model: GSC > QA-GNN > MHGRN > RoBERTa. This indicates that GSC has quite similar behaviors versus other GNN-based systems, and the reasoning capability of GSC is on par with existing GNN counterparts. This further reveals that counting plays an essential role in knowledge-aware reasoning for QA.

To verify our observations, we handcraft a hard edge counting feature feeding into a simple two-layer MLP for ablation study. As shown in Table 8, this hard counting model with 2-hop edge feature could

| Model | IHdev-Acc. (%) | IHtest-Acc. (%) |
|---|---|---|
| MLP + Counter (1-hop) | 78.02 (±0.05) | 73.62 (±0.12) |
| MLP + Counter (2-hop) | 78.30 (±0.09) | 74.13 (±0.08) |
| GSC w/ QA nodes | 79.11 (±0.22) | 74.48 (±0.41) |
| w/ 32 nodes | 78.52 (±0.58) | 74.40 (±0.25) |
| w/ 64 nodes | 78.53 (±0.77) | 73.93 (±1.09) |
| w/ 128 nodes | 78.50 (±0.96) | 72.78 (±1.15) |
| w/ 256 nodes | 78.32 (±0.60) | 73.89 (±0.63) |

Table 8: Ablation study on the hard counter with MLP (upper) and the maximum number of retrieved nodes (bottom), showing that 1) the hard counter achieves performance on par with GNN-based methods; 2) GSC works well even when only using entities occurred in QA context, which typically contains less than 32 nodes.

achieve a comparable performance of GSC, which even outperforms other GNN baselines. These impressive results not only show that our GSC is effective, but also prove that counting is an essential functionality in the process of the current knowledge-aware QA systems.

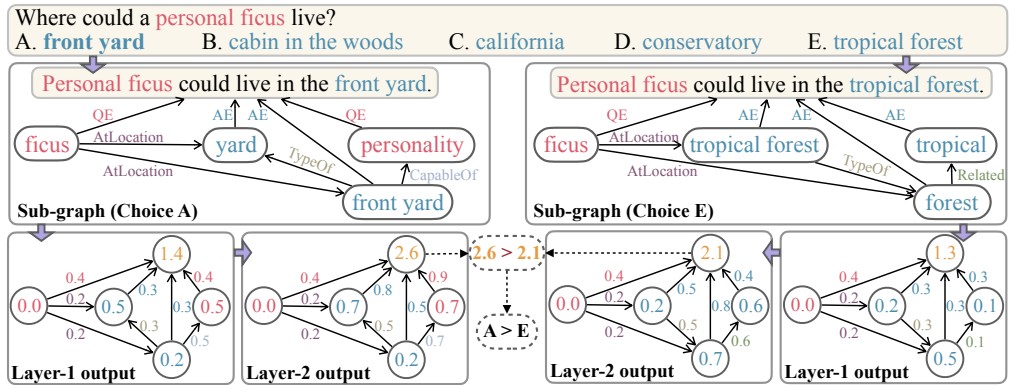

Figure 6: Our GSC is highly interpretable. For the retrieved sub-graph of each answer choice, we can directly observe the behavior of the model by print out the output edge/node values of each layer, so that we can trace back to see how the model scores the answer.

As shown in Figure 6, for the retrieved sub-graph of each answer choice, we can directly observe the behavior of the GSC by printing out the output edge/node values (soft count) of each layer. In this way, we can trace back to see why the model scores the answers like that. We list the runtime soft counts of GSC's edge encoder in Appendix A.4. A higher soft count means that the edge/node is more important, and can contribute more to the final graph score. This demonstrates the advantage of GSC as an interpretable reasoning module. In addition, GSC only reasons over the edges in the graph so that the concept embedding is not used, which makes our model better generalize to graph with new entities.

## 5 RELATED WORK

**KG-powered QA.** For KG-powered QA, traditional methods use semantic parsers to map the question to logical form (Berant et al., 2013; Yih et al., 2015) or executable program on KG (Liang et al., 2016), which typically require domain-specific rules and fine-grained annotations, making them difficult to handle the noise in questions and KG. To address these challenges, various deep neural models have been applied to KG-powered QA, such as memory networks (Kumar et al., 2015; Sukhbaatar et al., 2015; Miller et al., 2016), neural programmer for knowledge table QA (Neelakantan et al., 2016), neural knowledge-aware reader (Xiong et al., 2019; Sun et al., 2019), etc.

**GNN for KG-powered QA.** To further improve the neural reasoning capability, recent studies have explored applying GNNs to KG-powered QA, where GNNs naturally fit the graph-structured knowledge and show prominent results. KagNet (Lin et al., 2019) proposes GCN-LSTM-HPA for path-based relational graph representation. MHGRN (Feng et al., 2020) extend Relation Networks (Santoro et al., 2017) to multi-hop relation scope and unifies both path-based models and RGCN Schlichtkrull et al. (2018) to enhance the interpretability and the scalability. QA-GNN (Yasunaga et al., 2021) proposes a LM+GAT framework to joint reasoning over language and KG. We do not use conventional GNNs and our GSC is extremely simple and efficient even without parameters inside the GSC layers. KG embeddings are generally used in these systems. Zhang et al. (2018) employs propagation of KG embeddings to perform multi-hop reasoning. KagNet pre-trains KG Embedding using TransE initialized with GloVe embeddings. Recently, MHGRN and QA-GNN are proposed, which leverage pre-trained LM to generate embeddings for KG. In addition, QA-GNN also scores KG node relevance using LM as extra node embedding. We follow the data pre-processing procedure of QA-GNN and MHGRN, but our model does not use any KG embedding or node score embedding.

## 6 CONCLUSION

We investigate state-of-the-art GNN-based QA systems, and discover that they are over-parameterized and over-complex. Our diagnostic analysis using SparseVD shows that the initial node embeddings and some GNN layers are completely dispensable. Furthermore, our work reveals that GNN essentially works as a counter in the QA reasoning process. To verify this point, we design soft / hard counter models, which achieve comparable or even better experimental results than existing GNN-based methods. Our work is more of an explorative and investigational research, which points out how far we go in the area of knowledge-powered QA and provides helpful insights for the QA community to enlighten future research.

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

# A APPENDIX

## A.1 MULTI-HOP QA EXPERIMENTS

To apply our GSC to the task of multi-hop QA, we use a popular dataset MetaQA (Zhang et al., 2018), which contains more than 400k single and multi-hop (up to 3-hop) questions in the domain of movies. The questions were constructed using the knowledge base provided by the WikiMovies (Miller et al., 2016) dataset. We use the "vanilla" version of the QA pairs in the dataset, and we use the KG provided by the paper which consists of about 43k entities and 135k triples.

We do not encode the entities information in GSC, since our node embedding is initialized with zero. The edges in the KG are encoded with concatenated one-hot vectors according to the edge type (9 types of relation) and the node type (question/answer entity). Since the task of multi-hop QA does not provide answer choices, thus there is no context node inside the retrieved knowledge graph. We then adapt 3 units of GSC to align with this scenario:

**Edge encoder** We equip each GSC layer with a independent edge encoder to learn edge information for different hop. The input of the edge encoder is also adapted. Since various of questions can be asked to the same question entity, which means the retrieved graph for different question may be identical, we concatenate the LM encoded question text representation vector with edge type vector as the input of the edge encoder. So that the model could learn different edges embedding for different question text. Except the edge encoder is changed, we keep other parts of or GSC architecture the same.

| Methods | 1-hop | 2-hop | 3-hop |
|---|---|---|---|
| KV-Mem (Miller et al., 2016) | 96.2 | 82.7 | 48.9 |
| VRN (Zhang et al., 2018) | 97.5 | 89.9 | 62.5 |
| GraftNet (Sun et al., 2018) | 97.0 | 94.8 | **77.7** |
| GSC (ours) | **97.6** | **99.8** | 76.8 |

Table 9: Our GSC achieves comparable performance with the baselines on the MetaQA dataset, which indicates that our observations and hypotheses keep consistent multi-hop QA.

**Output mechanism** Since all the retrieved nodes can be the answer entities, we output all the node values as scores and the highest scored entity is our final answer. In the hit@1 setting, if our final answer is in the set of all the answer labels, we mark it correct.

**Supervision** Since there could be multiple correct answers in the KG given a question, we process the labels for each question as a multi-hot vector and use MSE loss instead of Cross Entropy as the supervision to the node scores.

We compared our GSC with three standard baselines KV-Mem (Miller et al., 2016), VRN (Zhang et al., 2018) and GraftNet (Sun et al., 2018) on MetaQA and report the hit@1 accuracy in Table 9. We can see that our GSC achieves comparable performance with the baselines on the MetaQA dataset, which indicates that our observations and hypotheses keep consistent multi-hop QA.

## A.2 HOW WE CONCLUDE THAT GNN IS A COUNTER

The process we conclude that GNN is a counter is not direct, our finding is based on sufficient experiments and obtained by a step by step process:

1. We find that the initial node embedding is dispensable and the layers inside the GNN preserve relatively low sparse ratio via the SparseVD analysis.

2. So we remove the initial node embedding layers and try to set different hidden dimension for GNN layers to see how the number of parameters affect the performance. The experiments result can be found at Table 10, where we do ablation study on the hidden dimension size of the message passing layers in the GAT

| Methods | IHdev-Acc. (%) | IHtest-Acc. (%) |
|---|---|---|
| GAT w/ hidden=2 | 77.32 (±0.62) | 74.08 (±0.12) |
| w/ hidden=4 | 77.07 (±0.08) | 74.30 (±0.42) |
| w/ hidden=8 | 77.12 (±0.33) | 74.65 (±0.33) |
| w/ hidden=16 | 77.01 (±0.21) | 74.59 (±0.68) |
| w/ hidden=32 | 77.26 (±0.19) | 74.62 (±0.43) |
| w/ hidden=64 | 77.37 (±0.43) | 73.97 (±0.29) |
| w/ hidden=128 | 77.40 (±0.38) | 74.21 (±0.37) |
| w/ hidden=256 | 77.56 (±0.30) | 74.27 (±0.61) |
| w/ hidden=512 | 76.80 (±0.54) | 73.35 (±0.70) |
| w/ hidden=1024 | 75.87 (±0.50) | 71.34 (±1.09) |

Table 10: Ablation study on the hidden dimension size of the GAT model, where we find that the model still works when the hidden size is extremely small, and the model get worse when the hidden size is extremely large due to the problem of over-fitting.

model, where we find that the model still works when the hidden size is extremely small, and the

model get worse when the hidden size is extremely large due to the problem of over-fitting. And we keep the hidden size of edge encoder with 32 since we find if the hidden size of edge encoder is extremely small the model will dis-converge.

3. Since the hidden dimension size of the message passing layers in the GAT model can not be set to even smaller, we design our own model with hidden dimension equal to 1 and simplify the attention based message passing layers to a non-parameterized operation. We surprisingly find that this model can still work and even work better. So we further study this model and find it works like a counter, so we name it Graph Soft Counter.

4. To further verify our perspective that the GNN works like a Counter, we design the hard counter setting, and find that it also works and could achieve comparable performance.

5. We may conclude that current GNNs in QA tasks work like a counter and counting for edges in graph are crucial in reasoning.

### A.3 SPARSE RATIO CURVE

We attach the details of the training curve of SparseVD dissection for QA-GNN as following. We find the linear query and key layers inside GNN layers and graph pooler generally can be compressed to a relatively low sparse ratio and this is more obvious at the fist three GNN layers. This indicated the these layers may be over-parameterized and the attention mechanism (Vaswani et al., 2017; Veličković et al., 2018) may degenerate to linear function in this case if we remove these layers.

As expected, the final fully connected layer (FC) preserved the highest sparse ratio, since it serves for directly generating the overall score of a choice and it is very efficient (the output dimension is 1). Except that, the edge encoder with a two-layer MLP has very high sparse ratio, which can be explained as the edge encoding is essential for GNN reasoning in these tasks. This is also why we only keep the edge encoder with relatively more learnable parameters and simplify the other components in GNN to design our Graph Soft Counter.

### A.4 SOFT COUNT OF EDGES

We list the top-30 edge triplets with highest soft counts that encoded by the edge encoder of our GSC as following. These combinations of edge types and node types have relatively higher counts, which means they can contribute more to the final graph score.

### A.5 SPARSEVD

To take a deep scope of the graph neural networks based knowledge-aware systems, we introduce a neural model pruning method Sparse Variational Dropout (SparseVD, (Molchanov et al., 2017)) into this scenario as a dissection tool to automatically dissect the graph network architecture. We implemented the SparseVD optimization based layer refer to the PyTorch code[*] released by the author. To keep the dissection in strict accordance with the theoretical derivation of Molchanov et al. (2017), we apply regulation to all the linear layers in the GNN modules. We attach the SparseVD problem formulation as well as the optimization target as following.

Consider a dataset $\mathcal{D}$ which is constructed from $N$ pairs of objects $(x_n, y_n)_{n=1}^N$. Our goal is to tune the parameters $w$ of a model $p(y \,|\, x, w)$ that predicts $y$ given $x$ and $w$. In Bayesian Learning we usually have some prior knowledge about weights $w$, which is expressed in terms of a prior distribution $p(w)$. After data $\mathcal{D}$ arrives, this prior distribution is transformed into a posterior distribution $p(w \,|\, \mathcal{D}) = p(\mathcal{D} \,|\, w)p(w)/p(\mathcal{D})$. This process is called *Bayesian Inference*. Computing posterior distribution using the Bayes rule usually involves computation of intractable multidimensional integrals, so we need to use approximation techniques.

One of such techniques is *Variational Inference*. In this approach the posterior distribution $p(w \,|\, \mathcal{D})$ is approximated by a parametric distribution $q_\phi(w)$. The quality of this approximation is measured in terms of the Kullback-Leibler divergence $D_{KL}(q_\phi(w) \,\|\, p(w \,|\, \mathcal{D}))$. The optimal value of variational

---

[*]https://colab.research.google.com/github/bayesgroup/deepbayes-2019/blob/master/seminars/day6/SparseVD-solution.ipynb

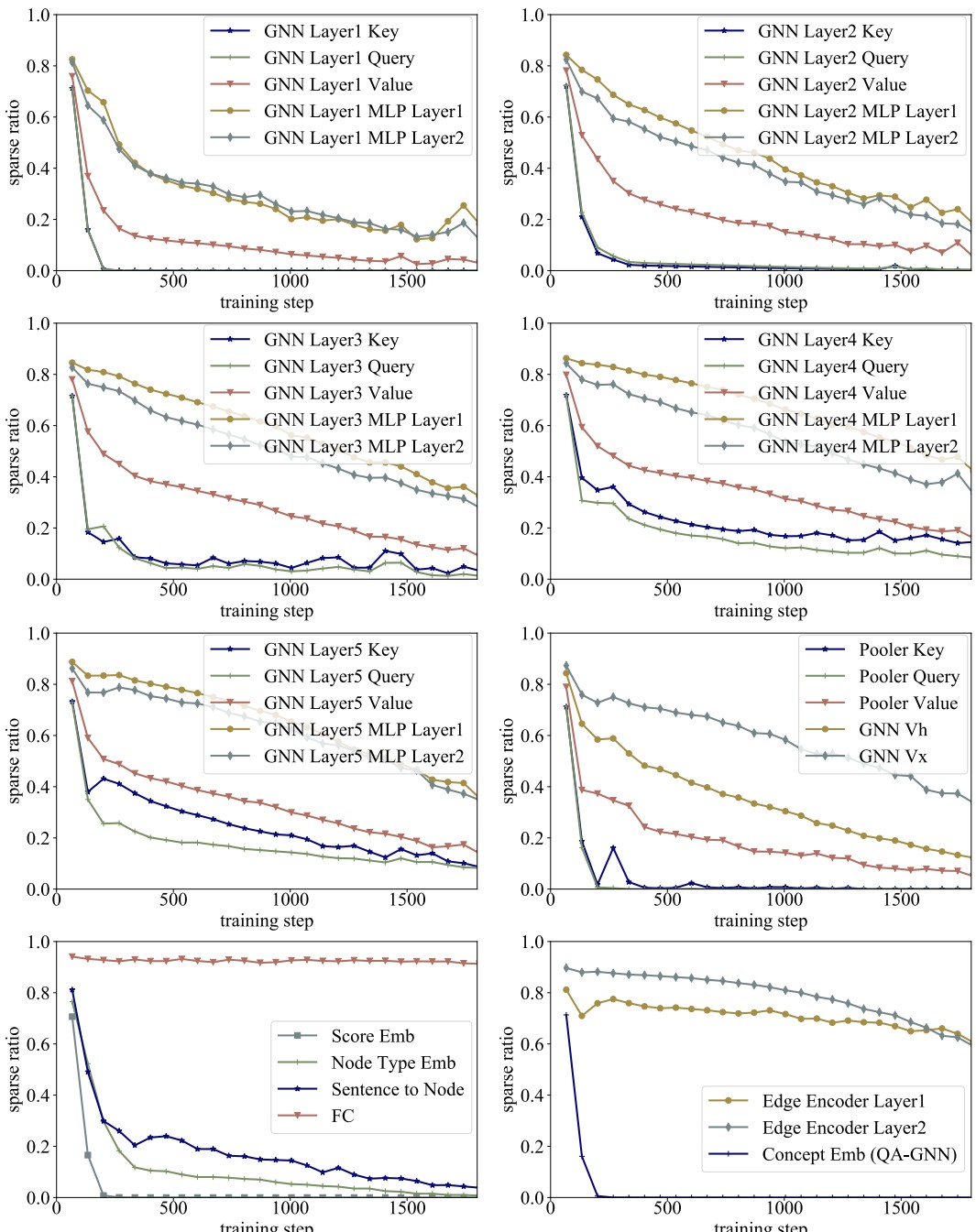

Figure 7: The sparse ratio curve when do SparseVD training for the QA-GNN systems.

parameters $\phi$ can be found by maximization of the *variational lower bound*:

$$\mathcal{L}(\phi) = L_{\mathcal{D}}(\phi) - D_{KL}(q_\phi(w) \,\|\, p(w)) \to \max_{\phi \in \Phi} \tag{1}$$

$$L_{\mathcal{D}}(\phi) = \sum_{n=1}^{N} \mathbb{E}_{q_\phi(w)}[\log p(y_n \,|\, x_n, w)] \tag{2}$$

It consists of two parts, the expected log-likelihood $L_{\mathcal{D}}(\phi)$ and the KL-divergence $D_{KL}(q_\phi(w) \,\|\, p(w))$, which acts as a regularization term.

| Head Node Type | Relation Type | Head Node Type | Soft Count Value |
|:---:|:---|:---:|:---:|
| Q entity | desires (inverse) | Q entity | 0.9632 |
| Q entity | desires | Q entity | 0.8902 |
| Q entity | has subevent (inverse) | Q entity | 0.8775 |
| Q entity | is | Q entity | 0.8470 |
| Q entity | has property | Q entity | 0.7344 |
| Q entity | does not desires | Q entity | 0.6995 |
| Q entity | does not desires (inverse) | Q entity | 0.6790 |
| Q entity | causes | Q entity | 0.6252 |
| Q entity | has property (inverse) | Q entity | 0.3531 |
| A entity | is | Q entity | 0.3073 |
| Q entity | causes (inverse) | Q entity | 0.2868 |
| Q entity | is part of | Q entity | 0.2758 |
| Q entity | is capable of (inverse) | Q entity | 0.2706 |
| A entity | has property | Q entity | 0.2424 |
| A entity | desires | Q entity | 0.2207 |
| Q entity | is the antonym of (inverse) | Q entity | 0.2137 |
| Q entity | is created by (inverse) | Q entity | 0.1720 |
| Q entity | is capable of | Q entity | 0.1681 |
| Q entity | is part of (inverse) | Q entity | 0.1581 |
| Q entity | desires (inverse) | A entity | 0.1310 |
| Q entity | is at location of | Q entity | 0.1006 |
| Q entity | is created by | Q entity | 0.0990 |
| Q entity | is at location of (inverse) | Q entity | 0.0914 |
| Q entity | is | A entity | 0.0737 |
| Q entity | has property | A entity | 0.0636 |
| Q entity | has subevent (inverse) | A entity | 0.0617 |
| O entity | is | Q entity | 0.0606 |
| O entity | has property | Q entity | 0.0512 |
| Q entity | desires | A entity | 0.0445 |
| Q entity | is the antonym of | Q entity | 0.0399 |

Table 11: We list the top-30 edge triplets with highest soft counts here, and it is generated by the edge encoder of GSC model. The combination of edge types and node types with a higher count means it can contribute more to the final graph score.

We formulate the model prunings (parsification) problem pruning problem as a variational inference problem and use the Kullback-Leibler Divergence approximation following (Achterhold et al., 2018; Molchanov et al., 2017) to constrain the model parameters to converge to the pruning prior: Original $-D_{KL}$ was obtained by averaging over $10^7$ samples of $\epsilon$ with less than $2 \times 10^{-3}$ variance of the estimation.

$$-D_{KL}(q(w_{ij} \,|\, \theta_{ij}, \alpha_{ij}) \,\|\, p(w_{ij})) \approx$$
$$\approx k_1 \sigma(k_2 + k_3 \log \alpha_{ij}) - 0.5 \log(1 + \alpha_{ij}^{-1}) + \mathrm{C} \tag{3}$$
$$k_1 = 0.63576 \quad k_2 = 1.87320 \quad k_3 = 1.48695$$

Where $\sigma(\cdot)$ denotes the sigmoid function. We can see that $-D_{KL}$ term increases with the growth of $\alpha$. It means that this regularization term favors large values of $\alpha$. The case of $\alpha_{ij} \to \infty$ corresponds

to a Binary Dropout rate $p_{ij} \to 1$ (recall $\alpha = \frac{p}{1-p}$). Intuitively it means that the corresponding weight is almost always dropped from the model. Therefore its value does not influence the model during the training phase and is put to zero during the testing phase.

We can also look at this situation from another angle. Infinitely large $\alpha_{ij}$ corresponds to infinitely large multiplicative noise in $w_{ij}$. It means that the value of this weight will be completely random and its magnitude will be unbounded. It will corrupt the model prediction and decrease the expected log likelihood. Therefore it is beneficial to put the corresponding weight $\theta_{ij}$ to zero in such a way that $\alpha_{ij}\theta_{ij}^2$ goes to zero as well. It means that $q(w_{ij} \,|\, \theta_{ij}, \alpha_{ij})$ is effectively a delta function, centered at zero $\delta(w_{ij})$.

$$\theta_{ij} \to 0, \quad \alpha_{ij}\theta_{ij}^2 \to 0$$
$$\Downarrow \tag{4}$$
$$q(w_{ij} \,|\, \theta_{ij}, \alpha_{ij}) \to \mathcal{N}(w_{ij} \,|\, 0, 0) = \delta(w_{ij})$$

In the case of linear regression this fact can be shown analytically. We denote a data matrix as $X^{N \times D}$ and $\alpha, \theta \in \mathbb{R}^D$. If $\alpha$ is fixed, the optimal value of $\theta$ can also be obtained in a closed form.

$$\theta = (X^\top X + \mathrm{diag}(X^\top X)\mathrm{diag}(\alpha))^{-1}X^\top y \tag{5}$$

Assume that $(X^\top X)_{ii} \neq 0$, so that $i$-th feature is not a constant zero. Then from (5) it follows that $\theta_i = \Theta(\alpha_i^{-1})$ when $\alpha_i \to +\infty$, so both $\theta_i$ and $\alpha_i\theta_i^2$ tend to 0.

In our dissection scenario, we optimize the weight $\theta_{ij}$to zero no longer for compressing the model but for figuring out which part of the model can be pruned out (sparse ratio to zero) without loss of accuracy, which indicates that part of model is not important and its output information is unused in optimization.

