# OpenReview forum: "GNN is a Counter? Revisiting GNN for Question Answering"
_ICLR.cc/2022/Conference — ICLR 2022 Poster_

### Official Review · Reviewer_zYDH · 2021-10-24

**Correctness:** 3
**Technical Novelty And Significance:** 4
**Empirical Novelty And Significance:** 4
**Recommendation:** 6
**Confidence:** 5

**Main Review:**

I feel this is a great paper that is somewhat "too good to be true". The proposed GSC achieves better performance than UnifiedQA on the OpenbookQA dataset but using 1/30 number of parameters (actually is this accurate? find the detailed comments below). The authors also perform several analysis on the proposed GSC, e.g., the number of parameters, ablation of GSC and so on.

My main question and concern is how/why GSC achieves such high performance with a somehow “disentangled” GNN+LM framework. By “disentangled”, I mean that the final qa_score is a sum of context_score (from LM) and graph_score (from GNN and KG). But it seems that the two terms have no interaction at all in GSC before the summation, especially for the calculation of graph_score. Besides the small number of parameters, I think this is the biggest difference between the paper and prior works such as QAGNN. QAGNN initializes the node embedding of the Context node as the Roberta embedding, and after several rounds of GNN and message passing, the information in the Roberta embedding is propagated to other nodes from the KG subgraph. However, in this work the graph_score is only calculated using the KG subgraph. Then the GSC will perform exactly the same with Roberta in theory under the following circumstances. Consider two different questions but with the same entities and same four answer candidates. Although the Roberta embedding of the question context will be different, the KG subgraph will be the same or really similar. Then the graph_score will also be similar. GSC will rely on context_score to really differentiate the reasoning process of the two questions. One such example being a question and its version with negation added.

Since the proposed GNN model is super simple (less than 10 lines as shown in Algorithm 1), I would strongly suggest the author open source their implementation during the review process and also for broader use of the method later.

- caption in table 2, remove “is” from “which makes our model size is extremely small”
- caption in table 7, are you sure unifiedQA and T5 are only 30x and 8x larger than your model? I feel these models are even larger.
- What is the difference between Learnable Param and Model size in Table 2? Why does GSC have the same param and model size?
- Figure 5 is interesting, it seems that even for a simple model like GSC, different random seeds still have a decent effect and the results still have around 8%-9% variance. Can the author explain the reason behind this?
- Is conceptnet the only knowledge graph the paper experimented with? I think it would make the paper much more impactful if the author can show that GSC works on another QA dataset and knowledge graph.
- Can it go beyond multiple choice question answering and find the answers over all the nodes on the KG?
- Can you give a detailed description on MLP+Counter in Table 8?
- Can you compare GSC with prior works on structured reasoning, e.g., negation?


**Summary Of The Paper:**

The paper designs a new GNN+LM model to do question answering and commonsense reasoning over knowledge graphs. The authors propose graph soft counter, a surprisingly simple and effective GNN module that counts edge mentions over the extracted subgraph for a question-answer pair. It has less than 1% trainable parameters compared with state-of-the-art GNN+LM models.



**Summary Of The Review:**

I am currently a little concerned about why the method achieves such high performance, but I would definitely increase my score after my concern/questions are addressed.

---

> ### Author Response · Authors · 2021-11-22
> **Our Response to Reviewer zYDH (Part 2)**
>
> 4. ___Learnable param and model size___
>
>      The learnable parameters mean the weights in GNN that update with gradients in the training, while the model size refers to the whole GNN model including the fixed pre-trained entity embedding with size 799273x1024 of the ConceptNet, which dominates the model size of QA-GNN/MHGRN. While our GSC does not use the pre-trained entity embedding, so we have the same param and model size.
>
> 5. ___Different random seeds result in a large variance___
>
>       We observed that this phenomenon also exists in the training phase, the prediction even has a large variance when the loss has converged to very small. We explain this phenomenon by the ambiguity and the noise of the data, we could see this point from the human baseline on the official leaderboard of CSQA/OBQA, where human has an 11.1% error rate on CSQA and 8.3% on OBQA. Since those data are labeled by humans, when the model is faced with these noisy data, the model is likely to pick one answer by random or some bias. So it appears to be a large variance in prediction.
>
> 6. ___Another QA dataset and knowledge graph___
>
>     In our paper, we've demonstrated the effectiveness of our GSC on two popular QA datasets, both of which are multiple-choice QA tasks. To further investigate the effectiveness of GSC on other forms of QA tasks, we pick up the challenging multi-hop QA and apply GSC to a popular multi-hop QA dataset MetaQA. In the updated Appendix A.1, we summarize the new experimental results (also attached below). Our GSC achieves comparable performance against other GNN-based methods on the MetaQA dataset, which indicates that our observations and hypotheses are consistent on the task of multi-hop QA.
>     |Methods| 1-hop | 2-hop | 3-hop |
>     |---|---|---|---|
>     |KV-Mem|  96.2%   |  82.7%   |  48.9%   |
>     | VRN |  97.5%   |  89.9%   |  62.5%   |
>     |GraftNet|  97.0%   |  94.8%   |  **77.7%**   |
>     | GSC(ours) |  **97.6%**   |  **99.8%**   |  76.8%   |
>
>
>     GSC achieves comparable performance against baselines on MetaQA.
>
>
> 7. ___Find the answers over all the nodes on the KG___
>
>     Yes, we are also curious about this idea, since the ConceptNet used by CSQA/OBQA is extremely large, so it is hard to apply it to all the nodes on the KG, but we actually do it on MetaQA since its KG is relatively small. In MetaQA’s multi-hop QA scenario, no answer choice is provided, so we have to score every node in KG. The detail of the setting and experiments can be seen in the Appendix A.1 of the paper.
>
> 8. ___Detailed description on MLP+Counter___
>
>     Here we simply use the counter function in Python to count the number of each type of edge, and save it as a fixed length vector (for all cobinations of edges), then we pass it to a MLP model to generate the score for the given QA pair. More detailed implementation can be directly seen in our open-source code. L229-243 of modeling/modeling_gsc.py is the MLP model and L208-253 of utils/data_utils.py is the hard counter.
>
> 9. ___Structured reasoning, e.g., negation___
>
>     Since our GSC on CSQA/OBQA works in a disentangle way (LM+GSC), different question texts with the same question entities have the same retrieved graph, so the GSC part can not handle the structure reasoning like negation, but the LM part may learn different representation from small structured changes in the text. So in our extra experiments on MetaQA which is exactly constructed for structured reasoning, we adapt the edge encoder in GSC to handle this problem. The detail of the setting and experiments can be seen in the Appendix A.1 of the paper.
>
> 10. ___Some typos in revision___
>
>     Thanks for your advice. We’ve corrected these typos mentioned by reviewers in our updated manuscript.

---

> > ### Comment · Reviewer_zYDH · 2021-12-04
> > **Thanks for the response**
> >
> > Thank you for the clarification. I have some followup questions.
> > (1) About the MetaQA experiment, how did you retrieve the subgraph of a given question? Is it a 1/2 hop subgraph of the entity in the question?
> > (2) For table 10, can you also try the performance with hidden dim=1?
> > (3) Since you have "upgraded" the edge encoder as a context-aware one, can you also run the experiment on CSQA and OBQA? A detailed analysis on the difference of the originally proposed encoder and the new edge encoder on handling different questions (e.g., negation) would be super helpful.

---

> > > ### Author Response · Authors · 2021-12-06
> > > **Our Response to follow-up questions (Part 2):**
> > >
> > >
> > > 3. ___Compare the context-aware edge encoder with the original one___
> > >
> > >     We adapt the context-aware edge encoder to the GSC on CSQA and OBQA. As shown in the following two tables, it achieves better performance on CSQA. While for CSQA, it just gets comparable performance. So it seems like the questions in CSQA are more context-aware. We further calculate the proportion of the questions with the negation word “not” in these two datasets. 4.6% of questions in CSQA contain “not” while this proportion for OBQA is only 0.1%, which supports the point that CSQA is more context-aware. In addition, for the questions with “ not ” in the dev split of CSQA, our context-aware GSC correctly answers 78.4% of them, while our original model only gets 72.5%. It may explain why the context-aware edge encoder outperforms the original one on CSQA.
> > >
> > >     ----
> > >
> > >     | Methods | IHdev-Acc | IHtest-Acc |
> > >     |:-----|--|-------|
> > >     |GSC(original)|79.11($\pm $0.22)|74.48($\pm $0.41)|
> > >     |GSC(context-aware)|**79.20**($\pm $0.25)|**74.94**($\pm $0.48)|
> > >
> > >     On CSQA, the context-aware edge encoder outperforms the original one.
> > >
> > >     ----
> > >
> > >     | Methods | RoBERTa-large | AristoRoBERTa |
> > >     |:-----|--|-------|
> > >     |GSC(original)|**70.33**($\pm $0.81)|86.67($\pm $0.46)|
> > >     |GSC(context-aware)|70.07($\pm $0.61)|**87.07**($\pm $0.95)|
> > >
> > >     On OBQA, the context-aware edge encoder achieves comparable performance.
> > >
> > >     ----
> > >
> > >     Moreover, we present predictions of two versions of GSC handling the questions with negation as follows. These 3 case studies of structured reasoning show how the context-aware GSC answers the questions with negation modifications.
> > >
> > >     ----
> > >
> > >     | Example (Original taken from CSQA Dev) |  original pred | context-aware pred |
> > >     |:-----|:--|:-------|
> > >     | [original 1] What has a shelf that does **not** allow you to see what is inside of it?   A. **chest of drawers**  B. stove C. hold alcohol D. bookcase E. grocery store  | D. bookcase ($\times$) | A. chest of drawers ($\checkmark$) |
> > >     | [negation flip 1] What has a shelf that does allow you to see what is inside of it?   A. chest of drawers  B. stove C. hold alcohol D. **bookcase** E. **grocery store**  | D. bookcase ($\checkmark$ just no change)  | E. grocery store ($\checkmark$ change to correct) |
> > >     | [original 2] What might happen if someone is **not** losing weight?  A. loose skin  B. beauty C. miss universe D. **death** E. healthier  | A. loose skin ($\times$) | D. death ($\checkmark$) |
> > >     | [negation flip 2] What might happen if someone is losing weight?  A. loose skin  B. beauty C. miss universe D. death E. **healthier**  | D. death ($\times$ change to incorrect) | E. healthier ($\checkmark$ change to correct) |
> > >     | [original 3] Where would you get a toothpick if you do **not** have any?  A. box  B. **grocery store** C. eyes D. chewing E. mouth  | B. grocery store ($\checkmark$) | B. grocery store ($\checkmark$) |
> > >     | [negation flip 3] Where would you get a toothpick if you do have any?  A. box  B. grocery store C. eyes D. chewing E. **mouth**  | B. grocery store ($\times$ just no change) | E. mouth ($\checkmark$ change to correct) |
> > >
> > >     ----
> > >
> > >     In the first example, the context-aware GSC answers the original question correctly while the original GSC does not. When we flip the negation, now both D and E are correct answers. The context-aware GSC changes to the new answer E, and the original GSC just keeps the same.
> > >
> > >     In the second example, the context-aware GSC answers the original question correctly while the original GSC does not. When we flip the negation, now E is the correct answer. The context-aware GSC changes to the new answer E, while the original GSC change to another wrong answer.
> > >
> > >     In the third example, both the context-aware and original GSC answer the original question correctly. When we flip the negation, now E is the correct answer. The context-aware GSC changes to the new answer E, while the original GSC just keeps the same.
> > >
> > >     To sum up, leveraging the context-aware encoder, the GSC’s ability to handle structured reasoning (e.g., negation) is enhanced.

---

> > > ### Author Response · Authors · 2021-12-06
> > > **Our Response to follow-up questions (Part 1)**
> > >
> > > Thanks for your additional comments and advice. We will elaborate on these points as follows.
> > >
> > >    ---
> > >
> > > 1. ___Subgraph of MetaQA___
> > >
> > >     For the k-hop (k=1,2,3) split, we retrieve the k-hop subgraph of the entity in the question, and we also use a k-layer GSC to process it since one layer corresponds to one hop.
> > >
> > >     ---
> > >
> > > 2. ___Hidden dim=1 for Table 10___
> > >
> > >     Yes, we can also put the result of dim=1 in Table 10 as follows, where it achieves comparable performance.
> > >
> > >     ---
> > >
> > >     | Hidden dim | IHdev-Acc | IHtest-Acc |
> > >     |:-----:|--|-------|
> > >     |1|77.78($\pm $0.41)|73.89($\pm $0.29)|
> > >     |2|77.32($\pm $0.62)|74.08($\pm $0.12)|
> > >     |8|77.12($\pm $0.33)|74.65($\pm $0.33)|
> > >     |32|77.26($\pm $0.19)|74.62($\pm $0.43)|
> > >     |128|77.40($\pm $0.38)|74.21($\pm $0.37)|
> > >     |512|76.80($\pm $0.54)|73.35($\pm $0.70)|
> > >     |1024|75.87($\pm $0.50)|71.34($\pm $1.09)|
> > >
> > >     ---
> > >
> > >     It should be mentioned that since we use the GAT implemented from QA-GNN, there are some dimension constraints for some layers (should >=2), such as the node score embedding layer:
> > >     ```
> > >     self.emb_score = nn.Linear(hidden_size//2, hidden_size//2)
> > >     ```
> > >     It will collapse if we directly set dim=1 so we have to remove these layers to make dim=1 works. Actually, how to make dim=1 works is also the starting point of our GSC model, and we find that the attention mechanism is also dispensable which results in the prototype of our final method.

---

> ### Author Response · Authors · 2021-11-22
> **Our Response to Reviewer zYDH (Part 1)**
>
> Thank you very much for the encouraging and constructive comments. We just open-sourced our code in an anonymous Github repo (https://github.com/anonymousGSC/graph-soft-counter) to help the research community further explore this interesting topic. We'll de-anonymize the repo after the review process.
>
> 1. ___“Disentangled” GNN+LM framework___
>
>     Actually, we are also surprised that the GSC works pretty well even without the question text representation, so we do extra ablation study including the hard counter and the interpretation to figure it out. Finally, we find that current GNNs on the dataset like CSQA/OBQA are mainly focusing on the correlation between the question entities and the answer entities, where the more connected edges between the answer and the question, the more possibly the answer is correct. Our GSC model serves as a soft scorer for the edges to assign different weights for different types of edges.
>      We further find that simply relying on the correlation doesn’t work on multi-hop QA datasets like MetaQA, since various questions can be asked to the same question entity, which means the graph retrieved from that entity for different questions may be identical. So we extend our GSC with a question-aware edge encoder, where we concatenate the LM encoded question text representation vector together with edge type vector as the input of the edge encoder. So that the GSC could learn different edges embedding for different question text, and it achieves comparable performance with the baselines on the MetaQA dataset. We put the detail of this part in Appendix A.1.
>
> 2. ___Open-source code___
>
>       Yes, as mentioned above, we just open-sourced our code in an anonymous Github repo (https://github.com/anonymousGSC/graph-soft-counter) to help the research community further explore this interesting topic. Thanks for your suggestion!
>
>
> 3. ___Compare model size with UnifiedQA and T5___
>
>     Yes, the numbers are even larger. We roughly compare our model size including GSC+RoBERTa with the T5-11B (UnifiedQA is also based on T5-11B). Since most of our model size is dominated by the Roberta-large and T5-11B is 33x and 9x larger than Roberta-large (334M), so these numbers apply to our model too.

---

> ### Author Response · Authors · 2021-12-03
> **Waiting for Additional Comments and Advice**
>
> Dear reviewer, thanks for your last reply, which is very helpful for our revision. And we are looking forward to your additional comments and advice.

---

### Official Review · Reviewer_pyaj · 2021-10-31

**Correctness:** 3
**Technical Novelty And Significance:** 2
**Empirical Novelty And Significance:** 2
**Recommendation:** 5
**Confidence:** 3

**Main Review:**

Strengths
- Interesting study about overly complex QA solutions that have emerged in recent times and hold the crown of being SOTA.
- A simple alternative to the complex GNN module for the QA systems which even beats the SOTA numbers in two benchmark datasets.

Weakness
-  The solution seems to be too good for the simplicity and the performance that it offers. It is hard to believe that such a simple model works so well. There ought to be better explanation/reasoning for one to convincingly accept the empirical findings of the paper about existing GNN.
- Not much novelty in the proposed solution.
- Writing can be improved to some extent.
- Few other concerns which I have stated below.

Main Review
- Overall, I liked the goal of the paper and line of attack. However, I am still not fully convinced with their finding that existing GNNs mostly do counting of edges, and counting the edges in the graph alone plays a crucial role in knowledge-aware reasoning.

- I am also not fully clear about the diagnostic tool called SparseVD. To make the paper self-contained, it's important to elaborate a bit more on SparseVD.

- Using SparseVD this paper finds that these GNN modules are over-parameterized: some layers in GNN can be pruned to a very low sparse ratio, and the initial node embeddings are dispensable. I agree with this finding but how does this imply that GNN modules are merely counting edges in the KG? This part is unclear to me. Especially, in Section 2.3, the paper looks at the plot of the sparse ratio of some representative layers during the SparseVD training of GNN reasoning modules. Based on these observations alone, the paper seems to conclude that GNN modules are merely counting edges in the KG? This seems to be purely an experiment-based hypothesis but I fail to grasp the underlying intuition.

- Moreover, this appears to be a somewhat straightforward application of SparseVD and the core novelty seems to be lacking.

- The description of the "Relevance Score" in Section 2.1 is quite unclear and confusing. A bit of rewriting is required here to bring out clarity. Particularly, the precise definition of the "context" here, and how it is retrieved by the masked LM.

- In Table 1 column titles, what is the meaning of “IH..”?

**Summary Of The Paper:**

In recent times, there has been a proliferation of QA systems that combine pre-trained Language Models (LMs) with Graph Neural Networks (GNNs) to solve QA tasks requiring human-like reasoning. The pre-trained language models facilitate access to the large knowledge which is implicitly coded into their parameters. On the other hand, GNN kind of performs reasoning over the explicit knowledge available in the form of KGs. Such systems have shown a significant performance improvement. The aim of this paper is to revisit these QA systems. The motivation behind this study is as follows. The paper starts with the premise that today’s QA systems have become more and more complicated and I fully agree with it. Therefore, this paper wants to revisit those systems and ask several basic questions: Whether GNN-based modules are under or over-complicated for QA?, What is the essential role they play in reasoning over knowledge? etc.

To answer these questions, this paper first analyzes SOTA GNN modules for QA and their reasoning capability. Further, this paper utilizes this analysis to design a simple yet effective graph-based neural counter that achieves improved QA performance on CommonsenseQA and OpenBookQA, two popular QA benchmark datasets which heavily rely on knowledge-aware reasoning.

These existing (LM + GNN) based approaches for QA systems follow a two-step paradigm: 1) schema graph grounding and 2) graph modeling for inference. In Step 1, a sub-graph of KG is retrieved that is related to the QA context and grounded on concepts. This graph is called a schema graph. In Step 2, graph modeling is carried out via an elaborately designed GNN module. This paper essentially analyzes SOTA GNN modules employed in Step 2 and their reasoning capability. For such an analysis, this paper employs SparseVD as a diagnostic tool. Using this tool, this paper finds that existing GNN modules used in QA systems are over-complicated for what they can accomplish in the QA reasoning process. This paper’s analysis reveals that knowledge-aware GNN modules may only carry out some simple reasoning such as counting and that counting of edges in the graph alone plays a crucial role in knowledge-aware reasoning. Based on these findings, this paper shows that even a simple counting model can achieve QA performance comparable to state-of-the-art GNN-based methods. Based on these insights, this paper proposes a Graph Soft Counter(GSC) which is a simple yet effective neural module as the replacement for existing complex GNN modules. The paper shows that with less than 1% trainable parameters compared to existing GNN modules for QA, the GSC module outperforms those complex GNN modules on two benchmark QA datasets. The hidden dimension of GSC layers is only 1, thus each edge/node only has a single number as the hidden embedding for graph-based aggregation. GSC is also claimed to be interpretable because the aggregation of 1-dimensional embedding can be viewed as soft counting of edge/node in graphs.


**Summary Of The Review:**

See my main review.

---

> ### Author Response · Authors · 2021-11-22
> **Our Response to Reviewer pyaj (Part 2)**
>
> 3. ___About SparseVD___
>
>     Thanks for your suggestion. To make our paper more clear and self-contained, we elaborate the details of SparseVD in Appendix A.6. SparseVD is a model compression method based on Variational Dropout Theory, which can be treated as a tool to prune the weights in the neural models. If the reader is interested, the official repo of SparseVD with video can be referred to via https://github.com/bayesgroup/variational-dropout-sparsifies-dnn.
>
>
> 4. ___How we conclude the GNN may just count edges___
>
>     We elaborate how we make the conclusion that GNN-based QA modules may just count edges in Appendix A.2. Specifically, in Table 10 (for your convenience, a simplified version is attached below), we do an ablation study on the hidden dimension size of the GAT model. We find that the model still works well even when the hidden size is extremely small, and the model gets worse when the hidden size is extremely large due to the problem of over-fitting. Then we further reduce the hidden size to 1, and design a hard counter model to verify our findings. After this step-by-step investigation, we come up with the conclusion that counting edges plays a crucial role in GNN-based QA modules.
>
>     | Hidden dim | IHdev-Acc | IHtest-Acc |
>     |:-----:|--|-------|
>     |2|77.32($\pm $0.62)|74.08($\pm $0.12)|
>      |8|77.12($\pm $0.33)|74.65($\pm $0.33)|
>     |32|77.26($\pm $0.19)|74.62($\pm $0.43)|
>     |128|77.40($\pm $0.38)|74.21($\pm $0.37)|
>      |512|76.80($\pm $0.54)|73.35($\pm $0.70)|
>      |1024|75.87($\pm $0.50)|71.34($\pm $1.09)|
>
>
>
>     GAT model is not sensitive with hidden size even it is extremely small, and an extremely large hidden size will do harm to the performance due to the overfitting.
>
>
>
> 5. ___The core novelty___
>
>     Our core novelty: We find that existing GNN-based QA modules are over-parameterized and over-complex. Our work reveals that GNN essentially works as a counter in the QA reasoning process. To verify this point, we design soft / hard counter models, which achieve comparable or even better experimental results than existing GNN-based methods. Our work is more of an explorative and investigational research, which points out how far we go in the area of knowledge-powered QA and provides helpful insights for the QA community to enlighten future research.
>
>
> 6. ___Clarify some items___
>
>     Thanks for your suggestion. We re-organized the "Relevance Score" part in Section 2.1 & 2.3 to make our paper more clear. The relevance score is introduced by QA-GNN to estimate the importance of KG nodes against the given QA context.
>
>     The name “IH..” denotes the in-house split (IHdev for in-house dev and IHtest for in-house test) setting for the CSQA dataset, which is proposed by MHGRN and also used by QA-GNN to efficiently evaluate the models since the official test is hidden. For clarity, we added the explanation of these terms in the caption of Table 1 & 4.

---

> > ### Comment · Reviewer_pyaj · 2021-11-29
> > **Thoughts after reading authors' feedback**
> >
> > Thank you so much for putting in hard work to address my concerns. Somehow, I am still not fully convinced with the answer to my question "why the counting of edges plays a crucial role?". Probably, it is something that requires further deep investigation. Regardless, I will be comfortable sticking to my original rating.

---

> > > ### Author Response · Authors · 2021-12-01
> > > **Our Response to Reviewer pyaj (about the counting of edges)**
> > >
> > > Thanks for your kind reply. We would like to add a bit of clarification of our statement that counting of edges plays a crucial role: In this work, we find that counting of edges is vital in the scope of GNNs for QA, and we didn't over-generalize this conclusion to other knowledge-aware reasoning scenarios, which is yet to be explored. Our finding is consistent on three popular benchmark datasets of both multiple-choice and multi-hop QA tasks. Specifically, we verify our finding in the following ways:
> > >
> > > - We experiment an extremely simple model of "MLP + edge type counting" on CommenseQA, which achieves comparable performance against GNN baselines. This clearly shows that the counting of edges is crucial for this task.
> > > - In our ablation study on GNN hidden dimension, we find that the performance can be preserved even if we reduce the dimension to just 1. Since the hidden dimension is 1, all the hidden activations are just single numbers. Thus, as illustrated in Figure 4, this 1-dimensional GNN turns into a counter to sum up the values over the graph.
> > > - We further examine how such a simple counting process achieves prominent QA performance by visualizing the internal values on nodes and edges inside our GSC. As illustrated in Figure 6, the counting process aggregates the relatively higher edge scores on the correct-answer graph, leading to the right choice.
> > >
> > > In addition, as mentioned by the reviewer [nabQ], the intrinsic property of widely used QA datasets may highlight the importance of edge counting, since those crowdsourced datasets contain questions with similar reasoning patterns. The repeated reasoning patterns appear to be specific paths of edge types in KG, thus simply counting of edges can perform pretty well on those QA datasets.
> > >
> > > If you still have any concerns or comments to help us improve the paper, please feel free to let us know. We sincerely appreciate your advice.

---

> ### Author Response · Authors · 2021-11-22
> **Our Response to Reviewer pyaj (Part 1)**
>
> Thank you very much for the constructive comments. We just open-sourced our code in an anonymous Github repo (https://github.com/anonymousGSC/graph-soft-counter) to help the research community further explore this interesting topic. We'll de-anonymize the repo after the review process.
>
> 1. ___Better explanation about how GSC works___
>
>     Actually, we are also surprised that the GSC works pretty well even without the question text representation, so we do extra ablation study including the hard counter and the interpretation to figure it out. Finally, we find that current GNNs on the dataset like CSQA/OBQA are mainly focusing on the correlation between the question entities and the answer entities, where the more connected edges between the answer and the question, the more possibly the answer is correct. Our GSC model serves as a soft scorer for the edges to assign different weights for different types of edges.
>      We further find that simply relying on the correlation doesn’t work on multi-hop QA datasets like MetaQA, since various questions can be asked to the same question entity, which means the graph retrieved from that entity for different questions may be identical. So we extend our GSC with a question-aware edge encoder, where we concatenate the LM encoded question text representation vector together with edge type vector as the input of the edge encoder. So that the GSC could learn different edges embedding for different question text, and it achieves comparable performance with the baselines on the MetaQA dataset. We put the detail of this part in Appendix A.1.
>      |Methods| 1-hop | 2-hop | 3-hop |
>     |---|---|---|---|
>     |KV-Mem|  96.2%   |  82.7%   |  48.9%   |
>     | VRN |  97.5%   |  89.9%   |  62.5%   |
>     |GraftNet|  97.0%   |  94.8%   |  **77.7%**   |
>     | GSC(ours) |  **97.6%**   |  **99.8%**   |  76.8%   |
>
>
>     GSC achieves comparable performance against baselines on MetaQA.
>
> 2. ___Why counting of edges plays a crucial role___
>
>     We discover the importance of edge counting in current GNNs for QA following a step-by-step research process, which is summarized below:
>     1) GNN layers preserve low sparse ratio;
>     2) We tried different hidden dimensions in GNN to reduce parameters;
>     3) The model still works when the hidden size is extremely small;
>     4) We designed a GNN with the hidden dimension equal to 1 which still works;
>     5) We surprisingly find that our model works like a counter so we name it GSC;
>     6) We further design a hard counter experiment to verify our hypothesis;
>     7) Counting is crucial for reasoning.

---

### Official Review · Reviewer_nabQ · 2021-11-02

**Correctness:** 3
**Technical Novelty And Significance:** 3
**Empirical Novelty And Significance:** 3
**Recommendation:** 8
**Confidence:** 3

**Main Review:**

Strengths:

- The paper is clearly written and easy to understand
- The analysis of the results done is exhaustive and insightful
- They get decent improvements in two multiple choice QA dataset, however it is currently unclear how their method will work for non multiple choice datasets.

Weaknesses /Questions for the authors

- The model seems to be designed specifically for multiple choice questions. A context node, all question entities and answer entities connected to them. Moreover the score from the graph is read off from a single node (which I believe is the context node? Line 6 of Algorithm 1). Would the same simple neural counter work for problems where answer choices are not available, for example graph classification problems? I think the paper will benefit a lot if this method is shown to be effective for a dataset which does not have multiple choice questions.
- The context node has all incoming edges and the mechanism in which the neural counter work all the weights of each edge will be added to the context node? Is there a correlation between how dense the context node is for a question and answer choice? For example, how does a simple baseline which predicts an option based on the number of incoming edges in the context node work?
- The output of the edge encoder (47 X 32 X 1) converts the edge into a scalar message which is essentially passed around. That part of the network does not take into account any representation of the given question. I am unsure why that is the case. Arent the edge weights supposed to be learn wrt the question. But where is the question involved in this? Is it only in the subgraph creation process. i.e. the subgraph of each question is created wrt the question and that is usually enough to learn meaningful edge weights?
- In Fig4, the node in the bottom right has a value of 0.3 at the start. However, if I understand correctly, nodes receive score only from incoming edges and nodes are initialzied with zero values. Then how does that node have a non-zero score at any iteration?
- In algorithm 1, is the qa_context generated for each question answer pair? If so, how does this generalize to a setting when the answer options are not available at all?
- In figure 6, why is the personality node missing from the graph in the right hand side? The personality node is gotten as a part of the question entity and hence should be common for all graphs
- Minor: The edge embedding is said to be 47. However, each edge is represented as [u_s, e_{st}, u_t], which makes it 2 * 4 + 38 = 46. Can you tell me where did I miss 1 dimension?

**Summary Of The Paper:**

For several knowledge intensive tasks such as question answering, it is common to use a mixture of language models (that encode knowledge implicitly in its parameters) with a graph neural network based model that can encode external knowledge from an knowledge graph (KG). This paper does an analysis of such GNN based QA models that do message passing over an external knowledge graph (KG) to gather external knowledge required to answer a question. They use sparse variational dropout to diagnose the contributions of various parts of the model and find that current GNN modules are over-complicated and over parameterized. Building upon their analysis, they design a simple 1-dimensional neural counter model that counts nodes/edges in the graph and find that they outperform existing complicated GNN architectures, suggesting that those complex architectures might just be performing some simple count-based operations.

In the design of their simple model they were able to remove all of the dense node embedding layer, represent edges as simple sparse vectors and messages were reduced to single numbers, thus achieving massing improvements in storage and efficiency.

**Summary Of The Review:**

The paper is definitely very interesting and is fun to read. I think it would benefit from experiments on datasets which are not multiple-choice.
Also, there are applications where the learned entity representations from GNNs are used for downstream tasks. Since this method is advocating for removing node embeddings, there should be a discussion added on how to handle those situations (Note: I personally think removing entity representations is a good idea as it has lot of advantages such as generalizing to new entities etc, but for completeness, I think a discussion regarding the same should be added)

======Update 11/26======
Most of my concerns have been addressed through the author rebuttal and I am changing my score to accept.

---

> ### Author Response · Authors · 2021-11-22
> **Our Response to Reviewer nabQ (Part 2)**
>
> 6. ___Question nodes in Figure 6___
>
>      Yes, there should be the same personality node in the graph since it is directly connected to the question. However, there is a large number of entities singly connected to only question or answer and we can not draw it all, so we just draw the nodes that have connections both with answer and question entities in Figure 6 for clarity.
>
> 7. ___Dimension of edge encoder___
>
>       Yes, the input dimension is (2*4+38=46), we’ve updated it in the revision. The reason why we write 47 before is that the prior works (QA-GNN, MHGRN) use the self-loop edge for every node in the graph, so there is one more edge type for them while we do not use the self-loop edge.
>
> 8. ___Removing node embeddings___
>
>     Yes, we advocate for removing node embeddings in our scenario since it has a lot of advantages such as generalizing to new entities. And we add a brief discussion in the revision to further elaborate on these merits.

---

> ### Author Response · Authors · 2021-11-22
> **Our Response to Reviewer nabQ (Part 1)**
>
> Thank you very much for the constructive comments. We just open-sourced our code in an anonymous Github repo (https://github.com/anonymousGSC/graph-soft-counter) to help the research community further explore this interesting topic. We'll de-anonymize the repo after the review process.
>
> 1. ___Multi-hop QA datasets (non multiple-choice)___
>
>      In our paper, we've demonstrated the effectiveness of our GSC on two popular QA datasets, both of which are multiple-choice QA tasks. To further investigate the effectiveness of GSC on other forms of QA tasks, we pick up the challenging multi-hop QA and apply GSC to a popular multi-hop QA dataset MetaQA. In the updated Appendix A.1, we summarize the new experimental results (also attached below). Our GSC achieves comparable performance against other GNN-based methods on the MetaQA dataset, which indicates that our observations and hypotheses are consistent on the task of multi-hop QA.
>
>     |Methods| 1-hop | 2-hop | 3-hop |
>     |---|---|---|---|
>     |KV-Mem|  96.2%   |  82.7%   |  48.9%   |
>     | VRN |  97.5%   |  89.9%   |  62.5%   |
>     |GraftNet|  97.0%   |  94.8%   |  **77.7%**   |
>     | GSC(ours) |  **97.6%**   |  **99.8%**   |  76.8%   |
>
>
>     GSC achieves comparable performance against baselines on MetaQA.
>
>
>
> 2. ___Simple baseline counting the number of incoming edges of context node___
>
>       Yes, that’s how we interpret why our method works. Our ablation study on MLP with a hard counter model (Table 8 upper) could explain this phenomenon: In MLP + Counter (1-hop) setting, we count the number of each type of edges in side the retrieved graph (only has context node + question entity node + answer entity node), which includes the incoming edges of the context node, and it also achieves a decent performance. This indicates that the GNN mostly relies on the correlation density of the context node and question/answer choice.
>
> 3. ___The question involved in Graph representation___
>
>       Actually, we are also supervised that the GSC works pretty well even without the question text representation, so we do extra ablation study including the hard counter and the interpretation to figure it out. Finally, we find that current GNNs on the dataset like CSQA/OBQA are mainly focusing on the correlation between the question entities and the answer entities, where the more connected edges between the answer and the question, the more possibly the answer is correct. Our GSC model serves as a soft scorer for the edges to assign different weights for different types of edges.
>
>
>     We further find simply relying on the correlation doesn’t work on Multi-hop QA datasets like MetaQA, since various questions can be asked to the same question entity, which means the graph retrieved from that entity for different questions may be identical. So we extend our GSC with a question-aware edge encoder, where we concatenate the LM encoded question text representation vector together with edge type vector as the input of the edge encoder. So that the GSC could learn different edges embedding for different question text, and it achieves comparable performance with the baselines on the MetaQA dataset. We put the detail of this part in Appendix A.1.
>
> 4. ___In Fig. 4, nodes have a non-zero score___
>
>       As we initialize the initial node embedding with zero, so the nodes’ input for the first GSC layer is all zeros. While the output non-zero values on the nodes will be the input for the next GSC layer. Hence, without loss of generality, we draw the second GSC layer with non-zero nodes input in Fig. 4.
>
> 5. ___Generalize to a setting without answer options___
>
>       In Appendix A.1, to apply our GSC to multi-hop QA, we use a popular dataset MetaQA. And our GSC achieves comparable performance with the baselines on the MetaQA dataset, which indicates that our observations and hypotheses keep consistent on multi-hop QA.

---

### Official Review · Reviewer_oUEP · 2021-11-03

**Correctness:** 4
**Technical Novelty And Significance:** 3
**Empirical Novelty And Significance:** 4
**Recommendation:** 8
**Confidence:** 4

**Main Review:**

Strengths:
1.	The analysis of existing GNN modules is interesting. The insights on dissection are novel and significant for building better reasoning modules.
2.	The proposed Graph Soft Counter (GSC) is simple but efficient and interpretable, achieving impressive QA performance on two popular benchmarks.
3.	The ablation study in Table 8 is helpful, which shows that even a simple hard counting model can achieve QA performance comparable to state-of-the-art GNN-based methods.

Weaknesses:
1.	The definition of multiple-choice question answering and some terms used in related work are not briefly introduced,  making it difficult to accurately understand some of the content. For example, “answer choice a \in C” in Section 2.1 is sudden, “relevance score” that “measure the quality of a path and prune the sub-graph”  in Section 2.1 is incomprehensible, “sparse ratio” in Section 2 is counterintuitive (intuitively, the higher sparse ratio means more weights to zero and a part of the model is less important), and “Vh” and “Vx” in Figure 7 are unknown.
2.	In Algorithm 1, the PyTorch-style code may be hard to read for some people, and the variable “inputs” is not stated.
3.	The statement “Our GSC do not use node embedding” in Table 2 is not rigorous. Because node values can be viewed as 1-dimensional node embeddings, and the variable “node_emb” in Algorithm 1 reflects this view.

Questions:
1.	Why is the input dimension of edge encoder 47? It is inconsistent with the description (2*4+38=46) that follows.
2.	In GSC, why initialize node values to 0 and scale edge values in the range of (0, 1)? Any intuitive explanation？
3.	The two datasets for the experiment mainly rely on common sense, and the complexity of reasoning is relatively low. Do the observations and hypotheses in the paper apply to other complex reasoning (e.g. multi-hop) QA datasets?


**Summary Of The Paper:**

The paper investigates several state-of-the-art GNN modules and finds that they are over-complicated: the initial node embeddings are dispensable and some layers are over-parameterized. Based on these observations, the paper designs Graph Soft Counter (GSC), a simple graph neural model which basically serves as a counter over the knowledge graph. Although GSC has less than 1% trainable parameters compared to existing GNN modules for QA, it outperforms state-of-the-art GNN counterparts on CommonsenseQA and OpenBookQA, two popular QA benchmark datasets which heavily rely on knowledge-aware reasoning. These experiments reveal that counting plays a basic and crucial role in reasoning and existing knowledge-aware GNN modules may only carry out some simple reasoning such as counting.

**Summary Of The Review:**

The observations are novel, the proposed model is simple but efficient, and the experimental results are impressive. However, the description of some preliminaries and implementation details is not clear enough.

---

> ### Author Response · Authors · 2021-11-22
> **Our Response to Reviewer oUEP**
>
> Thank you very much for the encouraging and constructive comments. We just open-sourced our code in an anonymous Github repo (https://github.com/anonymousGSC/graph-soft-counter) to help the research community further explore this interesting topic. We'll de-anonymize the repo after the review process.
>
> 1. ___Briefly introduction to some terms used in related work___
>
>     We have included more introduction to the definition of some terms in the current revision. We thank the reviewer for the suggestions.
>
> 2. ___Algorithm 1___
>
>     Thanks for pointing it out. Yes, there is a typo in Algorithm 1, the “inputs” should be “edge_emb” and we’ve updated it in the revision. In addition, we release our code in public to further help the reader understand our method. Algorithm 1 is corresponding to the file modeling/modeling_gsc.py, which is easy to read.
>
> 3. ___Node embeddings___
>
>     Yes, node values can be viewed as 1-dimensional node embeddings. We change “Our GSC do not use node embedding” into “Our GSC does not use initial node embeddings” since the initial node embeddings (e.g.,TransE) are not used in GSC.
>
> 4. ___Dimension of edge encoder___
>
>     Yes, the input dimension is (2*4+38=46), we’ve updated it in the revision. The reason why we write 47 before is that the prior works (QA-GNN, MHGRN) use the self-loop edge for every node in the graph, so there is one more edge type for them while we do not use the self-loop edge.
>
> 5. ___Scale edge values in the range of (0, 1)___
>
>     We scale the edge values to (0, 1) just for better visualization or interpretation. Actually, we find it still works if we change the sigmoid function of the edge encoder to ReLU or simply non-activation there, while the edge values with the arbitrary range are hard to interpret.
>
> 6. ___Multi-hop QA datasets___
>
>     In our paper, we've demonstrated the effectiveness of our GSC on two popular QA datasets, both of which are multiple-choice QA tasks. To further investigate the effectiveness of GSC on other forms of QA tasks, we pick up the challenging multi-hop QA and apply GSC to a popular multi-hop QA dataset MetaQA. In the updated Appendix A.1, we summarize the new experimental results (also attached below). Our GSC achieves comparable performance against other GNN-based methods on the MetaQA dataset, which indicates that our observations and hypotheses are consistent on the task of multi-hop QA.
>
>       |Methods| 1-hop | 2-hop | 3-hop |
>     |---|---|---|---|
>     |KV-Mem|  96.2%   |  82.7%   |  48.9%   |
>     | VRN |  97.5%   |  89.9%   |  62.5%   |
>     |GraftNet|  97.0%   |  94.8%   |  **77.7%**   |
>     | GSC(ours) |  **97.6%**   |  **99.8%**   |  76.8%   |
>
>
>     GSC achieves comparable performance against baselines on MetaQA.

---

### Author Response · Authors · 2021-11-22
**General Response: Revision Uploaded and Code Released**

We thank all reviewers for their comments. In addition to the specific response below, here we summarize our goal and the changes planned to be included in the revision. Specific changes include:

1. ___We released our code___

    We just open-sourced our code in an anonymous Github repo (https://github.com/anonymousGSC/graph-soft-counter) to help the research community further explore this interesting topic. We'll de-anonymize the repo after the review process.

2. ___Our core novelty___

    We find that existing GNN-based QA modules are over-parameterized and over-complex. Our work reveals that GNN essentially works as a counter in the QA reasoning process. To verify this point, we design soft / hard counter models, which achieve comparable or even better experimental results than existing GNN-based methods. Our work is more of an explorative and investigational research, which points out how far we go in the area of knowledge-powered QA and provides helpful insights for the QA community to enlighten future research.

3. ___Our GSC is also applicable to multi-hop QA___

    In our paper, we've demonstrated the effectiveness of our GSC on two popular QA datasets, both of which are multiple-choice QA tasks. To further investigate the effectiveness of GSC on other forms of QA tasks, we pick up the challenging multi-hop QA and apply GSC to a popular multi-hop QA dataset MetaQA. In the updated Appendix A.1, we summarize the new experimental results (also attached below). Our GSC achieves comparable performance against other GNN-based methods on the MetaQA dataset, which indicates that our observations and hypotheses are consistent on the task of multi-hop QA.


    |Methods| 1-hop | 2-hop | 3-hop |
    |---|---|---|---|
    |KV-Mem|  96.2%   |  82.7%   |  48.9%   |
    | VRN |  97.5%   |  89.9%   |  62.5%   |
    |GraftNet|  97.0%   |  94.8%   |  **77.7%**   |
    | GSC(ours) |  **97.6%**   |  **99.8%**   |  76.8%   |


    GSC achieves comparable performance against baselines on MetaQA.


4. ___How we conclude that GNN works like a counter in current QA tasks___

    In Appendix A.2, we added discussions on how we discover the conclusion that GNN works like a counter in QA tasks. Here we provide a step-by-step history of our research process leading to this conclusion:
   1) GNN layers preserve low sparse ratio;
   2) We tried different hidden dimensions in GNN to reduce parameters;
   3) The model still works when the hidden size is extremely small;
   4) We designed a GNN with the hidden dimension equal to 1 which still works;
   5) We surprisingly find that our model works like a counter so we name it GSC;
   6) We further design a hard counter experiment to verify our hypothesis;
   7) Counting is crucial for reasoning.

---

### Decision · Program_Chairs · 2022-01-20

**Decision:**

Accept (Poster)

**Comment:**

This paper proposes a graph soft counter (GSC) model which is very simple and lightweight  compared to the conventional graph neural network for solving QA tasks that benefit from knowledge graphs. Compared to the conventional KG-GNN combination, the proposed method is much simpler but produces better results for QA tasks. The paper originally dealt only with multiple-choice QA tasks, but during the rebuttal process, the authors added more complex QA tasks which the reviewers appreciated. Additionally, there was (and still remains) some concern over the exact reasons and mechanisms behind this "too good to be true" result, and the authors addressed this with additional ablation studies, to be included in the appendix. With the publicly released code, others will be able to try GSC and its too-good-to-be-true performance and figure out how it actually works.